# Using Simulated Pest Models and Biological Clustering Validation to Improve Zoning Methods in Site-Specific Pest Management

**Luis Josué Méndez-Vázquez** [1] , **Rodrigo Lasa-Covarrubias** [2] , **Sergio Cerdeira-Estrada** [3]
**and Andrés Lira-Noriega** [4,*]

[1] Red de Estudios Moleculares Avanzados, Instituto de Ecología A. C., Xalapa 91073, Mexico;
luis.mendez@posgrado.ecologia.edu.mx
[2] Red de Manejo Biorracional de Plagas y Vectores, Instituto de Ecología A. C., Xalapa 91073, Mexico;
rodrigo.lasa@inecol.mx
[3] Comisión Nacional para el Conocimiento y Uso de la Biodiversidad, Mexico City 14010, Mexico;
scerdeira@conabio.gob.mx
[4] CONACyT Research Fellow, Red de Estudios Moleculares Avanzados, Instituto de Ecología A. C.,
Xalapa 91073, Mexico
[*] Correspondence: andres.lira@inecol.mx

**Abstract:** Site-specific pest management (SSPM) is a component of precision agriculture that relies on spatially enabled agronomic data to facilitate pest control practices within management zones rather than whole fields. Recent integration of high-resolution environmental data, multivariate clustering algorithms, and species distribution modeling has facilitated the development of a novel approach to SSPM that bases zone delineation on environmentally independent subfield units with individual potential to host pest populations (eSSPM). Although the potential benefits of eSSPM are clear, methods currently described for its implementation still demand further evaluation. To offer clear insight into this matter, we used field-level environmental data from a Tahiti lime orchard and realistic simulations of six citrus pests to: (1) generate a series of virtual (i.e., controlled) infestation scenarios suitable for methodological testing purposes, (2) evaluate the utility of nested (i.e., within-cluster) partitioning essays to improve the accuracy of current eSSPM methods, and (3) implement two biological clustering validators to evaluate the performance of 10 clustering algorithms and choose appropriate numbers of management zones during field partitioning essays. Our results demonstrate that: (1) nested partitioning essays outperform zoning methods previously described in eSSPM, (2) more than one clustering algorithm tend to be necessary to generate field partition models that optimize site-specific pest control practices within crop fields, and (3) biological clustering validation is an essential addition to eSSPM zoning methods. Finally, the generated evidence was integrated into an improved workflow for within-field zone delineation with pest control purposes.

**Keywords:** algorithms; clustering; modeling; pest control; precision agriculture; site-specific; virtual pests

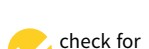



## 1. Introduction

Site-specific pest management (SSPM) is a component of precision agriculture (PA) that relies on spatially explicit agronomic data to facilitate pest control practices within homogeneous sub-field units (i.e., management zones or MZ) rather than whole fields [1–3]. Although the integration of precision inputs such as satellite imagery and climatic records into modern-day agriculture is relatively new, the first explorations of SSPM date back to the middle 1990s when data generated in the field of integrated pest management (IPM; e.g., pest samples, estimations of pest-induced crop damage) was geographically enabled by global positioning systems (GPS) and cutting edge variable rate technologies

(VRT; e.g., automated tractors, planters). Such a fusion of tools and concepts facilitated the consolidation of site-specific insect pest management (SSIPM) [3], a multidisciplinary approach to IPM which aims to partition infested crop fields into "treatment" and "no treatment" zones based on interpolated maps of within-field pest densities and economic thresholds of tolerance to pest-induced crop damages [4–7].

Recently, the integration of high-resolution environmental data, multivariate clustering techniques, and species distribution modeling (SDM) has led to the development of "ecological site-specific pest management" (eSSPM), an ecologically oriented approach to IPM which bases the delineation of MZ on environmentally independent sub-field units with individual potential to host pest populations [8]. SDM implements a variety of statistical mechanisms (i.e., modeling algorithms) to infer the spatial distribution of species based on correlations between their known geographic occurrences and the environmental conditions associated with them [9]. eSSPM field partitioning essays (i.e., delineation of MZ) are based on the following sequential steps: (1) description of cause-effect relationships between mapped environmental variables and within-field pest distributional patterns, (2) partitioning of a target crop field into a maximum number of MZ, (3) redefinition of MZ via SDM algorithms, (4) validation of environmental independence between MZ, and (5) classification of MZ based on their potential to host pest populations [8]. Although the prospective benefits of eSSPM are relevant and straightforward (e.g., controlled pesticide use, increase of crop value), field partition models generated by this approach are prone to show different sub-optimal results such as presence zones nested within absence clusters, presence zones insensitive to differentiated levels of pest infestation, more than one pest absence zones, and inaccurately delimited MZ [8].

Different factors explain eSSPM current limitations. First, the development of field partitioning essays based on single-time implementations of multivariate clustering (MC) algorithms, since pest absence zones within a crop field can consist of more than one environment equally unsuitable for pest establishment but still recognizable as independent MZ [8]. Second, the lack of clear-cut criteria to select appropriate MC algorithms during field partitioning essays, which is essential because the final topology of field partition models used in PA is highly influenced by the clustering approach used to compute them [10]. Third, the redefinition of sub-field units using SDM algorithms, due to SDM's tendency to generate zonal models with different degrees of spatial overlap between some MZ and incomplete representation of others [8]. Finally, the selection of optimal numbers of MZ based on measurements of environmental overlap between sub-field units (i.e., Schoener's *D*) rather than true clustering validation indexes (CVI). CVI are equations designed to evaluate the results of clustering analyses based on the degree of congruence between natural groups and the data used to create them (i.e., internal validation) or between natural groups and some other external reference (i.e., external validation) [10,11].

The development of this paper was based on two assumptions. First, to overcome the methodological limitations currently reported for eSSPM, field partitioning essays should consist of a two-steps process (i.e., nested field partitioning) where a rough distinction between pest presence and pest absence zones (i.e., binary field partitioning) precedes the subdivision of resulting sub-field units (i.e., complementary field partitioning). Second, external validation of eSSPM field partitioning essays based on biologically interpretable CVI should facilitate the selection of optimal MC algorithms to be used and appropriate numbers of MZ to be delineated. To prove these statements: (1) we used high-resolution environmental data from a Tahiti lime orchard and realistic simulations of six common citrus pests to generate a series of virtual infestation scenarios suitable for methodological testing purposes; (2) we implemented a series of nested field partitioning essays to test their capability to minimize sub-optimal zoning results reported for current eSSPM methods, and (3) we used two biologically meaningful CVI to compare the performance of 10 MC algorithms and to determine appropriate numbers of MZ to be considered during field partitioning essays. The use of simulated pest data allowed the development of testing essays under controlled virtual scenarios, an advised condition to assess modeling method-

ologies in ecology since it grants researchers unrestricted access to statistical processes and evidence necessary for drawing robust conclusions about natural mechanisms [12–14].

Five main contributions are presented in this paper. First, a clear distinction between SSIPM and eSSPM as conceptually and methodologically independent implementations of SSPM. Second, robust empirical evidence regarding the utility of nested field partitioning essays on overcoming the limitations reported for current eSSPM zoning approaches. Third, solid empirical evidence regarding the utility of biologically meaningful CVI to test the performance of MC algorithms and select adequate numbers of MZ during eSSPM zoning essays. Fourth, the first precedent on the simultaneous use of multiple MC algorithms to delineate MZ within the context of PA. Finally, an up-to-date workflow that considerably improves the accuracy of zoning methods presently implemented in eSSPM (Figure 1).

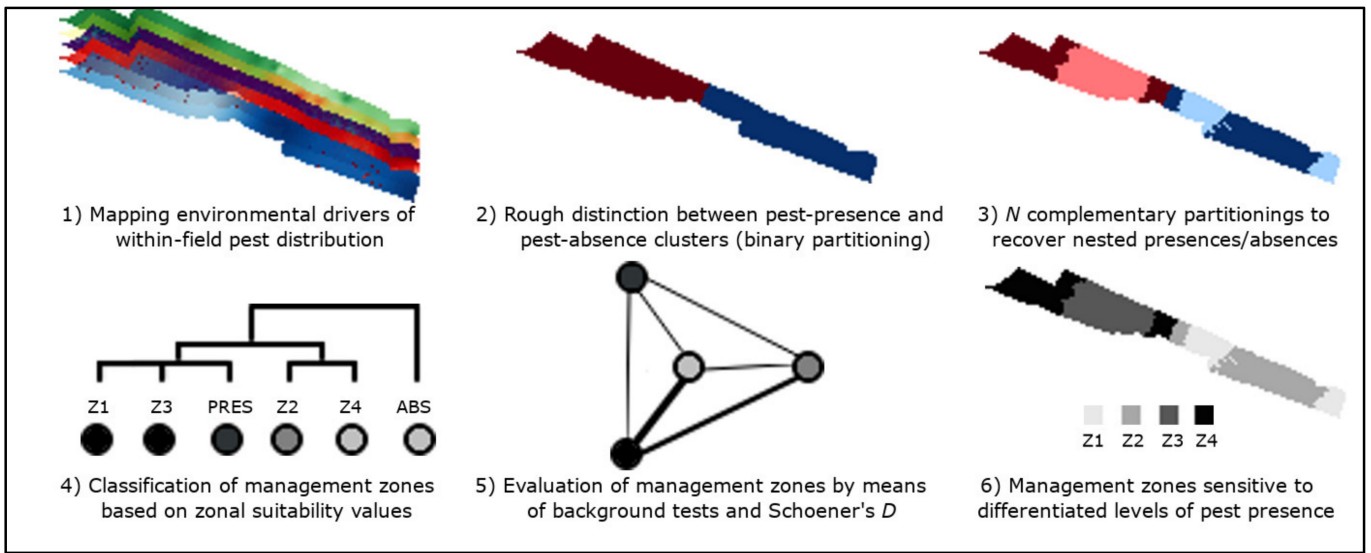

**Figure 1.** New workflow proposed to delineate management zones with pest control purposes.

## 2. Materials and Methods

### 2.1. Summary

The development of this work was based on the following methodological steps: (1) environmental representation of the experimental orchard by means of precision sampling tools (i.e., unmanned aerial vehicle, multispectral camera, data loggers, georeferenced soil samples, georeferenced pest samples), (2) probabilistic modelling of six virtual pests within the boundaries of the experimental orchard (i.e., phytopathogenic nematode, bacterial canker, fungal foot rot, insect-transmitted disease, invasive weed, phytophagous mite), (3) "binary field partitioning essays" to distinguish between pest presence and pest absence zones within the experimental orchard, (4) "complementary field partitioning essays" to distinguish differentiated levels of pest presence and to identify presence zones nested within absence clusters, (5) evaluation of field partition models by means of biologically meaningful CVI (i.e., biological homogeneity index or BHI and biological stability index or BSI), and (6) refinement of field partition models by means of hierarchical dendrograms (of environmental relationships between zones), bubble charts (of zonal suitability values) and visual networks (of zone environmental independence). The resulting observations were used to update zoning methods currently described in eSSPM.

### 2.2. Study Site

Environmental data were collected from a nine years old, artificially irrigated orchard (6.5 ha) dedicated to the commercial production of Tahiti lime (variety "Cucho," *Citrus aurantium* × *Citrus latifolia*, 6 m × 4 m between individuals) in the central region of Veracruz, Mexico. Climatic conditions associated with this region are warm humid, with

abundant summer rains and annual temperatures between 24 °C and 26 °C [15]. Predominant soil types range from sandy-clay to sandy-loam, with pH values ranging from 6.62 to 6.99 [16]. Surrounding landscapes are represented by agricultural plantations (e.g., sugar cane, corn, beans) and small remnants of dry forest, which was the primary ecosystem of the region before recent agricultural expansion [17]. The selection of this study area (i.e., Carrillo Puerto municipality) was based on its contribution to regional citrus activities and abundance of small commercial plantations. In contrast, the experimental orchard was chosen on account of its available historical data (i.e., environmental, production-related).

### 2.3. Data Sets

Four data sets were used to represent environmental conditions within the experimental orchard (1 m$^2$/pixel): multispectral aerial imagery, microclimatic data logs, presence-absence data of citrus pests, and georeferenced soil samples.

Multispectral imagery was captured using an unmanned aerial system (UAS) integrated by a low-cost quadcopter (3DR Solo, discontinued in 2019) and a cheap sports camera (Ekken 4K, 60 FPS) modified with a planar lens designed for vegetation analysis (i.e., NDVI-7; red-edge 750 nm, green 500–565 nm, blue 450–485 nm). Recent studies have used similar platforms to approach vegetation biophysical features [8,18,19]. The UAS was deployed once over the experimental orchard on October 17th of 2018 approximately at zenith (between 14:00 and 14:30 h. local time) to guarantee maximum radiation conditions and minimum shadow effects. This UAS followed a photogrammetric route designed to take images with 60% side overlap and 80% vertical overlap at a flight altitude of 50 meters above the takeoff site. A set of 12 fixed ground control points (e.g., georeferenced vinyl squares on the ground) was used to facilitate aerial image spatial referencing.

Microclimatic data was sampled using nine Arduino-based data loggers assembled and programmed in the Biogeography laboratory of the Instituto de Ecología, A.C. (IN-ECOL). Arduino is an open-source electronics prototyping platform based on simple, customizable hardware and software [20]. Data loggers were installed evenly across the orchard below fully grown trees, while ambient temperature and humidity sensors were placed 15 cm above the ground as in Méndez-Vázquez et al. [8]. Loggers were set to record information with a frequency of 60 minutes for 20 days, from October 28th to November 17th of the year 2018.

Georeferenced soil samples and presence-absence data of *Phytophthora* sp. "foot rot" and "brown rot" were collected from 73 randomly selected Tahiti lime trees. Two stages of foot rot were considered. Resinous wounds and callus tissue near the grafting area were associated with "active" and "inactive" foot rot infections, respectively [21]. After careful inspection of each tree, a soil sample of approximately 400 gr was collected from the uppermost 30 cm of the topsoil, where roots of citrus trees mainly develop [22]. Once in the laboratory, 100 gr of each available sample were used to prepare 1:5 dilutions in demineralized water as in Méndez-Vázquez et al. [8]. Such dilutions were used to measure soil pH and electrical conductivity (EC) values using a multipurpose sensor for monitoring water quality (YERYI TDS/EC/PH/TEMP meter).

### 2.4. Environmental Predictors of Virtual Pests

Multispectral images captured via UAS were used to generate three outputs: one multispectral orthomosaic (blue, green, and red edge bands), one digital surface model (DSM), and one digital terrain model (DTM). These products were created using Agisoft Photoscan (V1.2 for Debian Linux distributions), an image analysis software widely used in PA that facilitates the creation of maps from UAS imagery [23,24]. The native resolution of such outputs was re-scaled from 20 cm$^2$/pixel to 1 m$^2$/pixel to avoid computationally heavy processes and still achieve very high spatial resolution in our results. Four sub-products were obtained from the manipulation of spectral data. Red edge, green and blue bands of the orthomosaic were used to calculate a vegetation index highly correlated to plant metabolism and stress (i.e., single-band normalized differences vegetation index

or SI-NDVI [21]).　SI-NDVI (NDVI from now on) values facilitated the estimation of fractional vegetation cover (FVC) based on methods described by Thorp, Hunsaker, and French [25]. This measurement of plant area per surface unit is closely related to leaf area index and evapotranspiration [26,27]. The available DTM was used to compute maps of flow accumulation and topographic roughness index (TRI) using algorithms implemented in functions "r.terraflow" [28] and "r.tri" [29] of GRASS GIS 7 [30]. Within-field patterns of crop cover and terrain features are known environmental drivers of different agricultural pest species [27,31].

Microclimatic data logs were summarized into averages, maxima, and minima of every sampled variable (i.e., ambient temperature, relative humidity). Using logger coordinates and the inverse distance weights algorithm (IDW) implemented in the function "v.surf" of GRASS GIS 7, three raster maps were created to represent the experimental orchard in terms of average temperature, mean relative humidity, and vapor pressure deficit (VPD). This last variable is closely related to evapotranspiration and ecosystem function [32] and was calculated by implementing Allen's equation based on temperature and humidity data [33]. IDW is a statistical interpolation technique historically used to generate surface models of climatic variables and within-field pest distributions [34,35]. Although methods based on semivariograms (e.g., kriging) are a more standard approach to interpolate point-based data in PA [11,36,37], IDW is much simpler to implement, is less demanding in computing power, and is readily available in practically every GIS software today.　The resulting interpolated maps were not accurate representations of the environment associated with the experimental orchard but reductionist models that simplified the evaluation of complex environment-pest interactions in geographic space.　The influence exerted by climatic conditions over the distributional patterns of pests is well known [38–40].

Soil EC, soil pH and presence-absence data collected from citrus trees were also spatially interpolated using the IDW algorithm as in Corwin and Lesch [41] and Méndez-Vázquez et al. [8]. Soil pH and EC are relevant variables for agriculture due to their close relationship to crop productivity and soil physical properties, respectively [42,43]. Citrus foot rot and brown rot are different manifestations of *Phytophthora* sp. infections that affect crop productivity and facilitate the establishment of secondary diseases [44].

Twelve digital maps of environmental features were generated (in TIFF format). Variable names, codes, and methods used to compute them are presented in Table 1.

**Table 1.** Environmental predictors, their corresponding codes, and the estimation methods used to compute them.

| Code | Variable | Estimation Method |
|---|---|---|
| aFRot | active citrus foot rot | IDW interpolation of presence-absence data |
| flowAccum | flow accumulation | "r.terraflow" function of GRASS GIS 7 |
| cropFVC | fractional vegetation cover | $FVC = (1 + NDVI)/(1 - NDVI) \times NDVI\hat{}0.5$ |
| iFRot | inactive citrus foot rot | IDW interpolation of presence-absence data |
| relHum | mean relative humidity | IDW interpolation of data logs |
| sunRad | mean sub-canopy radiation | IDW interpolation of data logs |
| cropNDVI | single image NDVI | $SI\text{-}NDVI = (NIR - BLUE)/(NIR + BLUE)$ |
| soilEC | soil electrical conductivity | IDW interpolation of soil samples |
| soiPH | soil pH | IDW interpolation of soil samples |
| TRI | topographic roughness index | "r.tri function" of GRASS GIS 7 |
| VPD | vapor-pressure deficit | $VPD = esm - ea$ |

IDM: All IDW interpolation essays were executed using the "v.surf" function of GRASS GIS 7. esm: esm = (esmn + esmx)/2; esmn = 0.6108 × exp((17.27 × min Temp)/(min Temp + 273.3)); esmx = 0.6108 × exp((17.27 × max Temp)/(max Temp + 273.3)). ea = (mean RH/100) × esm.

## 2.5. Within-Field Distribution of Virtual Pests

Six pairs of uncorrelated predictor variables were used to simulate known distributional patterns of six virtual pests within the experimental orchard. Distributional maps of each pest are presented in Figure 2, whereas specific environmental ranges considered dur-



ing pest design are shown in Table 2. Pest virtualization was performed using R statistical software's "virtualspecies" package [45].

**Figure 2.** Distributional patterns of virtual pests (1–6) simulated within the experimental orchard. Values close to 1 (dark colors) represent regions of higher pest suitability.

**Table 2.** Environmental predictors, response functions, and parameterization values used to design virtual pests.

| Pest | Variable 1 | Fun. Var 1 | Range Var 1 | Variable 2 | Fun. Var 2 | Range Var 2 |
|------|-----------|-----------|-------------|-----------|-----------|-------------|
| 1 | VPD | normal | m = 0.55, sd = 0.25 | TRI | normal | m = 0.2, sd = 0.15 |
| 2 | flowDir | quadratic | a = 3, b = 1, c = 0.25 | sunRad | custom | m = 195, diff = 55, prob = 0.95 |
| 3 | relHum | quadratic | a = 3, b = 1, c = 0.25 | aFRot | logistic | beta = 0.3, alpha = 0.25 |
| 4 | soilPH | logistic | beta = 10, alpha = 1 | cropNDVI | normal | m = 0.05, sd = 0.1 |
| 5 | cropHeight | normal | m = 1.5, sd = 0.1 | ambTemp | quadratic | a = 3, b = 1, c = 0.25 |
| 6 | iFRot | logistic | beta = 0.75, alpha = 0.05 | soilEC | normal | m = 155, sd = 35 |

Distributional patterns of pest 1 were driven by temperature–humidity interactions (i.e., vapor pressure) and terrain features (i.e., topographic roughness) known to facilitate the proliferation of phytopathogenic nematodes specialized in citrus crops (i.e., *Tylenchulus semipenetrans*) [46].

Pest 2 responded to the existence of places prone to flooding (i.e., direction of flow accumulations) and low exposition to sunlight (i.e., sun radiation), where canker-producing bacteria (i.e., *Xanthomonas axonopodis*) can survive for days [47,48]. Pest 3 was inspired by fungal diseases (i.e., *Phytophthora* sp. foot rot/brown rot) that become active during the most humid months of the year (i.e., relative humidity) [21]. The distribution of pest 4 was based on an insect-transmitted disease (i.e., citrus greening) that proliferates better on citrus trees (i.e., NDVI) already exposed to physiological stress (i.e., pH) [49]. Pest 5 mimicked a generic undesired weed that invades bare soil areas of citrus orchards (i.e., FVC) when warm microclimates occur (i.e., ambient temperature). Finally, distributional patterns of pest 6 were based on those of a phytophagous mite (i.e., white/broad mite, *Polyphagotarsonemus latus*) whose populations thrive on trees damaged by previous diseases (i.e., inactive *Phytophthora* sp. foot rot) and highly stressing environmental conditions (i.e., electrical conductivity) [21].

### 2.6. Nested Field Partitioning Essays

As mentioned before, the field partitioning approach implemented by current eSSPM methods shows relevant shortcomings during the delineation of MZ, such as presence zones nested within absence clusters, presence zones insensitive to differentiated levels of pest infestation, more than one pest absence zones, and inaccurately delimitated MZ [8]. To avoid these scenarios, MZ delineation essays implemented in this work were based on a two-step approach that we denominated nested field partitions. The first step consisted of binary field partitions where 500 random and spatially independent points representative of the experimental orchard (in terms of environmental factors relevant to the distribution of each simulated pest) were used to develop clustering essays that facilitated a rough distinction between pest presence and pest absence zones.

The second step consisted of complementary field partitions where 500 representations of presence-only and absence-only zones were used to implement within-cluster field partitions useful to identify differentiated levels of pest infestation and nested pest presence/absence zones. The main difference between binary and complementary field partitioning essays is that the former aims to partition the target crop field into two sub-field units (i.e., pest presence and pest absence clusters). In contrast, the latter seeks to partition binary sub-field units into several MZ that facilitate whether the "rescue" of nested presence/absence zones or the recognition of differentiated levels of pest infestation.

Ten MC algorithms were used to partition the experimental orchard binarily or complementarily (Table 3). These algorithms were implemented using R statistical software's "clValid" package [50] and were selected on account of their known capability to group biological data sets [10,50]. A more profound explanation of such clustering approaches is presented in Appendix A.

**Table 3.** Multivariate clustering (MC) algorithms compared during field partitioning essays developed in this work.

| Method | Acronym | Class | Reference | Package |
|---|---|---|---|---|
| Average linkage | AL | hierarchical | [51] | fastcluster |
| Clustering large applications | CLA | partitioning | [52] | cluster |
| Complete linkage | CL | hierarchical | [51] | fastcluster |
| Divisive analysis | DIA | hierarchical | [52] | cluster |
| Fuzzy analysis | FNY | partitioning | [52] | cluster |
| Model-based clustering | MCL | model-based | [53] | mclust |
| Partitioning around medioids | PAM | partitioning | [52] | cluster |
| Self-organizing maps | SOM | machine learning | [54] | kohonen |
| Single linkage | SL | hierarchical | [51] | fastcluster |
| Ward's linkage | WL | hierarchical | [55] | fastcluster |

Since it was not always possible to generate "perfect" field partition models (i.e., containing completely homogeneous clusters), binary partitions of the experimental crop field yielded one of four possible scenarios: (1) the partition model included clusters that clearly distinguished between pest presence and pest absence zones, (2) the model included one accurate absence cluster and a presence cluster that incorrectly hosted absence zones, (3) the model included one accurate presence cluster and an absence cluster that incorrectly hosted presence zones, and (4) both clusters in the model included a mixture of presence and absence zones. Complementary partitioning essays were implemented on presence-only clusters for scenarios (1) and (2). This facilitated the differentiation of pest levels (scenario 1) and the isolation of nested pest absence zones (scenario 2), depending on the case. Scenarios (3) and (4) demanded the partitioning of both pest presence and pest absence clusters, which facilitated the isolation of nested pest presence zones (scenario 3) and the distinction between pest presence and pest absence zones (scenario 4).

### 2.7. Validation of Field Partition Models

Selection of best field partition models (i.e., testing the performance of MC algorithms) and appropriate numbers of MZ were based on BHI and BSI indexes (0-1). BHI is an external measure for genetic clustering validation proposed by Datta and Datta [56] that determines how homogeneous clusters in a partition model are (higher values meaning a higher homogeneity) in terms of biologically meaningful categories called "functional classes" (i.e., genetic functions). In our case, surrogate environmental classes (e.g., high presence, low presence, pest absence) were generated by applying different suitability thresholds to distribution maps representative of the simulated pests. The specific number of presence levels and thresholds used to define them varied according to the case. The package "clValid" calculates BHI based on the following equation:

$$BHI(C, B) = \frac{1}{K} \sum_{k=1}^{K} \frac{1}{n_k(n_k - 1)} \sum_{i \neq j \in C_k} I(B(i) = B(j)), \tag{1}$$

where $n_k$ equals $n(C_k \cap B)$, which is the number of annotated categories (i.e., pest levels) in statistical cluster $C_k$, $B(i)$ is the functional class containing category $i$, and $B(j)$ is the functional class containing category $j$.

A second evaluation based on BSI measures was performed in cases where more than one field partition model shared the highest BHI values. BSI tests clustering consistency for observations with similar biological functionality (higher values meaning higher stability) [50]. To do so, new clustering essays are developed by removing one sample at a time (from the clustered data set) and cluster membership of observations with similar functional annotation is compared with cluster memberships observed during essays based on all available samples. BSI can also be calculated by the "clValid" package through the following equation:

$$BSI(C, B) = \frac{1}{F} \sum_{k=1}^{F} \frac{1}{n(B_k)(n(B_k) - 1)M} \sum_{l=1}^{M} \sum_{i \neq j \in B_k} \frac{n\left(C^{i,0} \cap C^{j,l}\right)}{n\left(C^{i,0}\right)}, \tag{2}$$

where $F$ is the total number of functional classes, $C^{i,0}$ is the statistical cluster containing observation $i$, and $C^{j,l}$ is the statistical cluster containing observation $j$ when column $l$ is removed.

After the best partition models were selected (i.e., MC algorithms and number of MZ that maximized BHI/BSI values), cluster membership numbers were interpolated within the experimental orchard using the IDW algorithm included in GRASS GIS 7 (function "v.surf"). Maps resulting from these interpolations were reclassified to eliminate decimal values and produce management zones containing unique cluster membership numbers.

### 2.8. Classification of Management Zones

After binary and complementary field partition models were fused to generate preliminary field partition models, management zones were categorically classified (e.g., absence, low presence, high presence) based on a decision support system consisting of hierarchical dendrograms, bubble charts, and cartographic projections.

Individual dendrograms were generated by hierarchically clustering (i.e., AL) environmental values representative of management zones included in a preliminary field partition model and true presence/absence zones known to operate within the experimental orchard. True presence and absence zones were delineated by selecting a presence-absence suitability threshold (PAST) for each virtual pest and reclassifying all values in their distribution models. All suitability values below the PAST established for a given pest were reclassified to 0, whereas those equal or above such a PAST were reclassified to 1. The resulting dendrograms were used to represent the existing relationships between MZ included in

preliminary field partition models and the environmental closeness of such MZ to true presence/absence zones delimited for their corresponding pests.

Bubble charts were computed based on the mean suitability values observed within MZ included in preliminary field partition models. Zonal suitability averages were calculated by overlaying the distribution model generated for a target pest (see "Within field distribution of virtual pests") and MZ included in its corresponding partition model. Bubble size and color corresponded with their represented values (bigger/darker bubbles meant higher suitability values).

Cartographic representations of preliminary field partition models and the known distribution of their corresponding pests facilitated the interpretation of hierarchical dendrograms and bubble charts previously described.

### 2.9. Validation of Management Zones

According to PA theory, after a target crop field has been partitioned into *n* sub-field units, the appropriateness of MZ needs to be evaluated to determine whether there are real differences between them (or not) in terms of the agricultural phenomenon to be managed (e.g., soil properties, yield). Historically, this task has been accomplished by implementing strategies as simple as ANOVA models or as complex as mixed linear models (MLM). However, no standard method has been described to this date [10,11].

Since in our case MZ are expected to show individual potential to host pest populations, their represented environments are also likely to be differentiated from one another. This condition can be evaluated with SDM background tests, which are tools initially designed to measure the level of environmental overlap between SDM models generated for two species (pairwise comparisons) using Schoener's *D*. This index (0–1) is sensitive to ecological similarities (between geographic entities) given by diet and microhabitat variables [57,58].

In this work, the generation of zonal SDM models and the implementation of background tests was based on the "ENMTools" package for R [59], which estimates the spatial distribution of compared species based on the Maxent (i.e., maximum entropy) algorithm [60] and estimates Schoener's *D* with the following equation:

$$D = 1 - \frac{1}{2}\left(\sum_{ij}|Z_{1ij} - Z_{2ij}|\right), \tag{3}$$

where $Z_{1ij}$ and $Z_{2ij}$ represent the occupancy of entities 1 and 2, respectively.

In practice, environmental and geographic samples (i.e., environmental values, geographic coordinates; *n* = 500) of management zones included in preliminary field partition models were used to implement pairwise background tests that generated individual matrices of between-zone overlaps (one for each pest). Such matrices were used to feed a set of visual networks that represented management zones as labeled nodes, the environmental similarity between zones as numbers next to each link (i.e., *D*), and the statistical significance of a particular nexus value as the link's width. In cases where preliminary field partition models considered more than one absence zones, these were fused into a single absence cluster before MZ networks were computed.

The threshold used to determine similarities between MZ was an environmental overlap equal to or greater than 10% ($D \geq 0.1$). Regardless of the observed similarity between zones, statistically significant environmental relationships ($\alpha = 0.05$) were represented as "thick" links between nodes. In contrast, statistically insignificant ones were drawn as "slim" (low similarity values below the alpha level) and "normal" (high similarity values below the alpha level) links between nodes.

The results of this exercise (i.e., environmental relationships between MZ) were used to generate a series of final field partition models that were regarded as the best possible options to facilitate pest management practices within the experimental orchard from an eSSPM perspective.

### 3. Results

#### 3.1. Nested Field Partitioning Essays (Binary)

All BHI and BSI values used to evaluate binary partition models (roughly distinguish between pest presence and pest absence clusters) are presented in Figure 3. These results show that the best performing MC algorithms for pests 1 to 6 were, respectively: CL, MCL, SL, CL, SL, and WL (Figure 4). Binary field partition models that displayed BHI values approaching 1 showed excellent capability to distinguish between pest presence and pest absence zones (i.e., pest 5, pest 6). The exception to this pattern was observed in pest 2 which generated a field partition model with a BHI value of 0.93 but was unable to distinguish accurately between pest presence and pest absence zones on one half of the experimental orchard (i.e., south). It is worth noticing that the BSI value displayed by the binary partition model generated for pest 2 (i.e., 0.69) was significantly lower than BSI values observed in partition models developed for pests 5 and 6 (0.97 and 0.89 respectively).

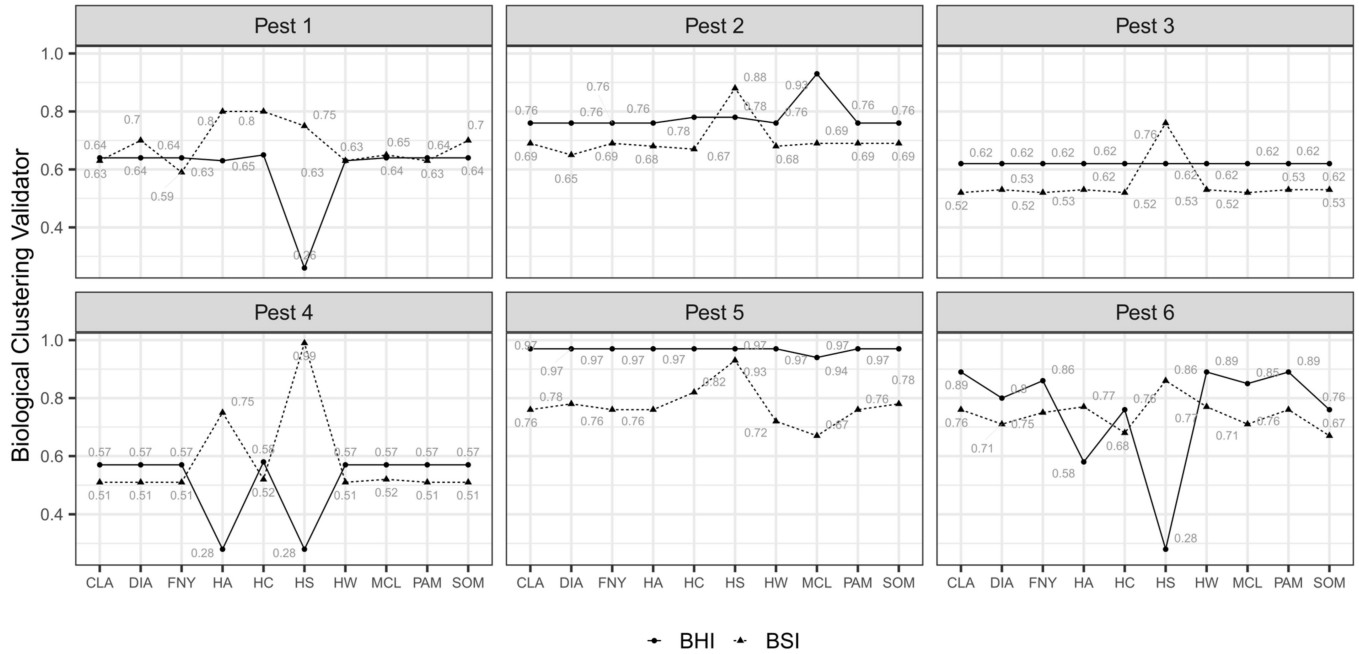

**Figure 3.** Biological homogeneity index (BHI) and biological suitability index (BSI) values calculated for binary partitions modeled by the compared algorithms. These indexes base partition selection on the highest observed values.

#### 3.2. Nested Field Partitioning Essays (Complementary, Presence-Only)

Best performing MC algorithms during field partitioning essays developed within presence-only clusters are shown in Figure 5 (BHI) and Figure 6 (BSI). For pests 1 to 6 best performing algorithms were: SOM, SL, CL, MCL, DIA, and MCL (Figure 7). In this case, partitioning essays developed over homogeneous geographic entities (i.e., BHI approaching 1) were prone to recognize highly homogeneous zones (pest 2, pest 4). An exception to this pattern was observed in pest 6, where partitioning of a homogeneous presence-only cluster resulted in poorly homogeneous zones (BHI: 0.67).

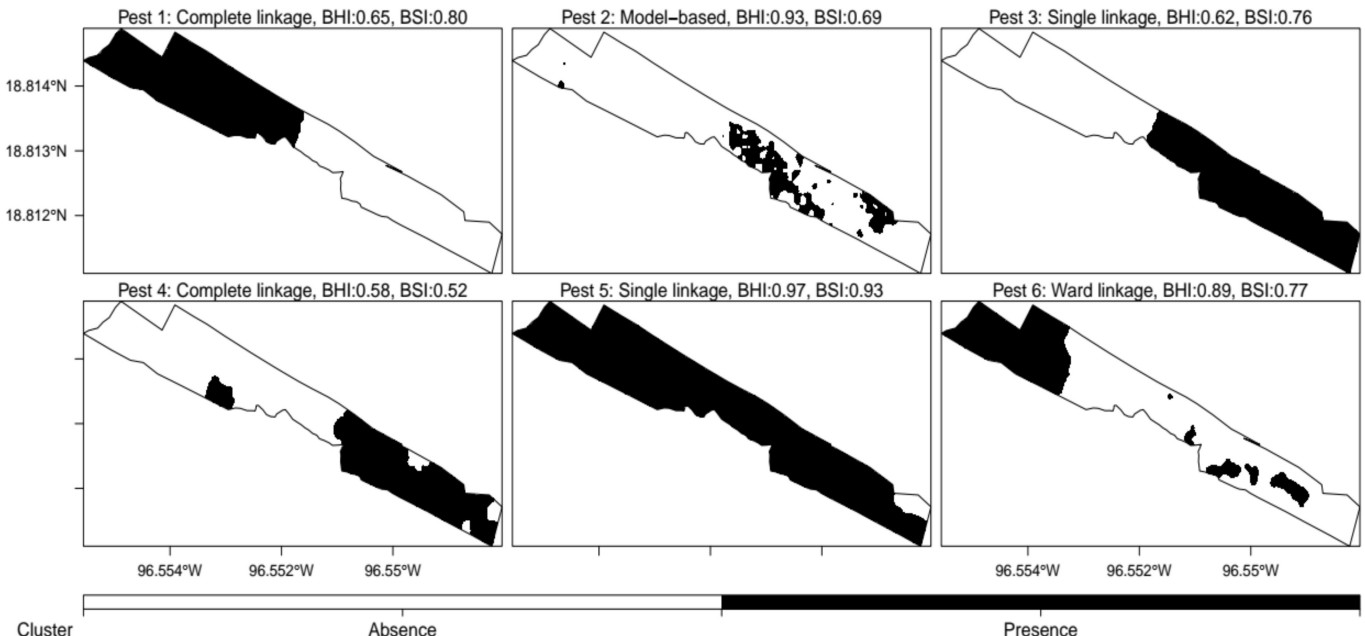

**Figure 4.** Best binary partition models generated for the six virtual pests simulated within the experimental orchard.

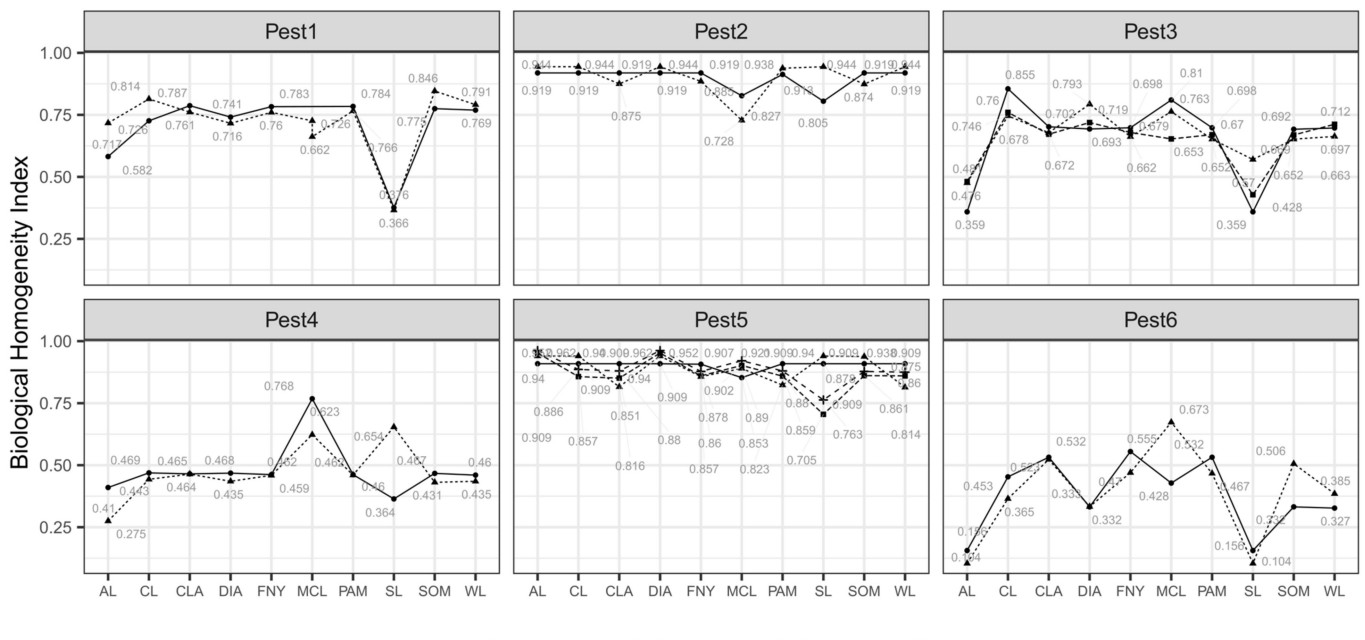

**Figure 5.** Biological homogeneity index (BHI) values calculated for complementary partition essays implemented over presence-only clusters of binary models. This index bases partition selection on the highest observed values.

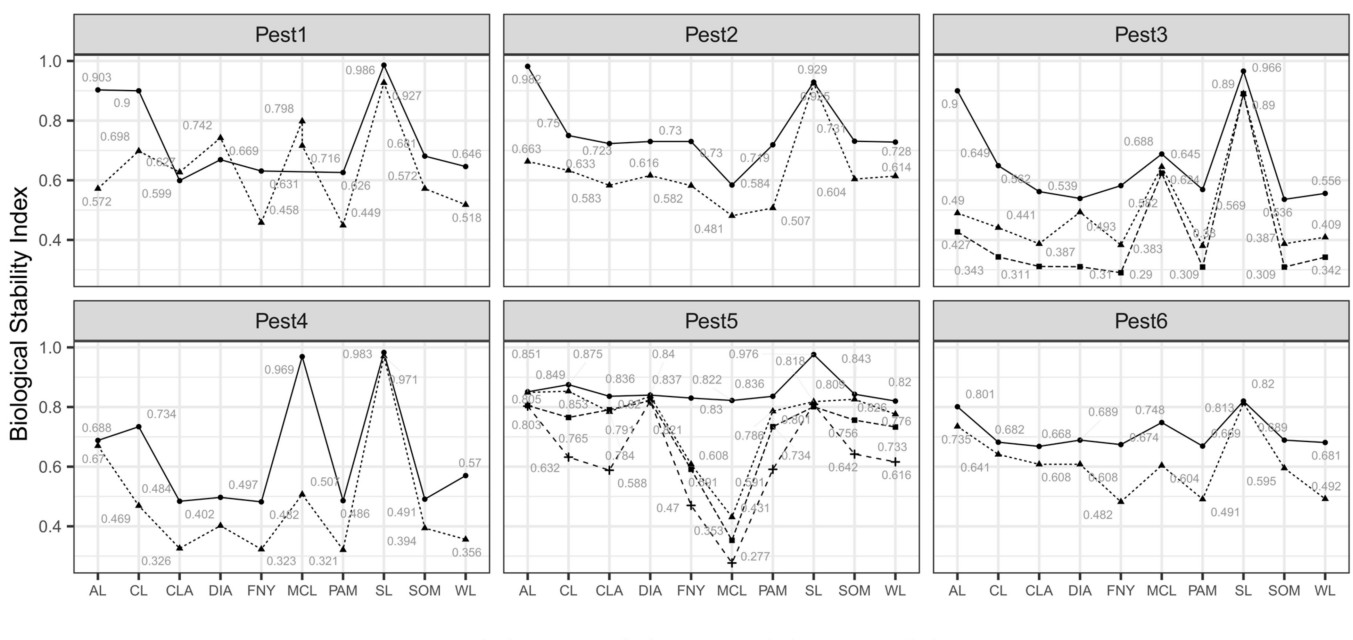

**Figure 6.** Biological suitability index (BSI) values calculated for complementary partition essays implemented over presence-only clusters of binary models. This index bases partition selection on the highest observed values.

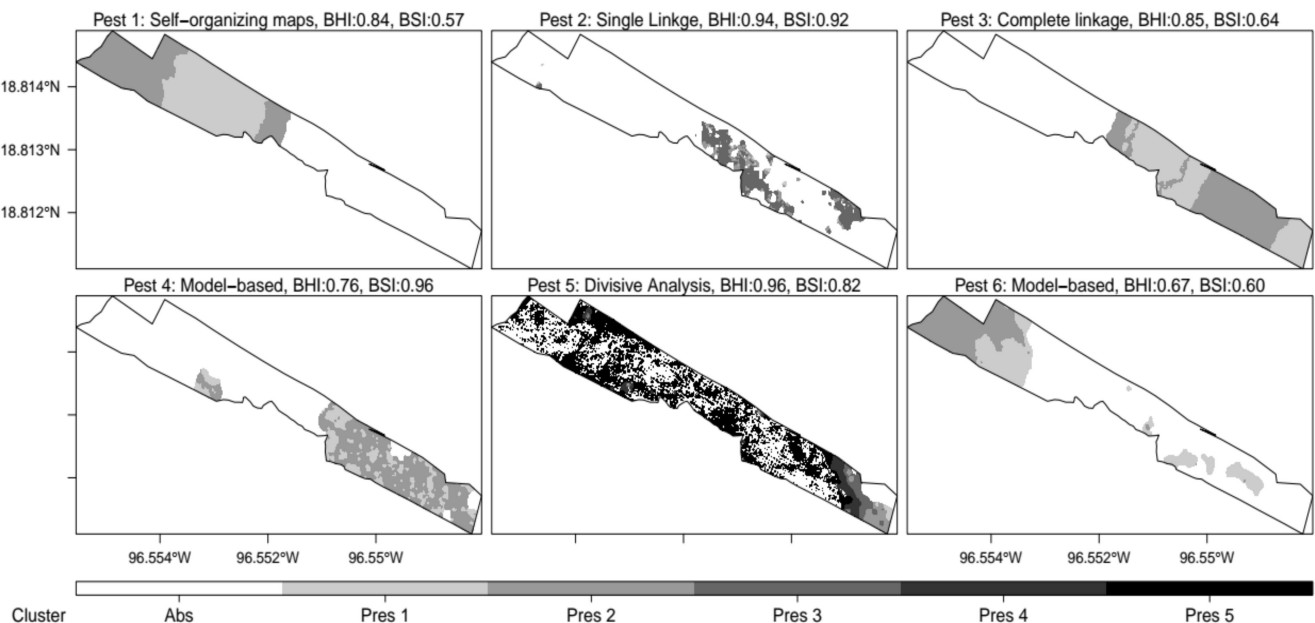

**Figure 7.** Best partition models of presence-only clusters used to identify differentiated levels of pest presence and isolated nested absences within the experimental orchard.

### 3.3. Nested Field Partitioning Essays (Complementary, Absence-Only)

Implementation of field partitioning essays within absence-only clusters was pertinent only for binary partitions of pests 1 to 4. Figures 8 and 9 show that best performing MC algorithms for these pests were: WL, SL, MCL and MCL (Figure 10). In this case, all generated field partition models displayed relatively high BHI values, even those that were developed within poorly homogeneous absence-only zones (pests 1, 3, and 4).

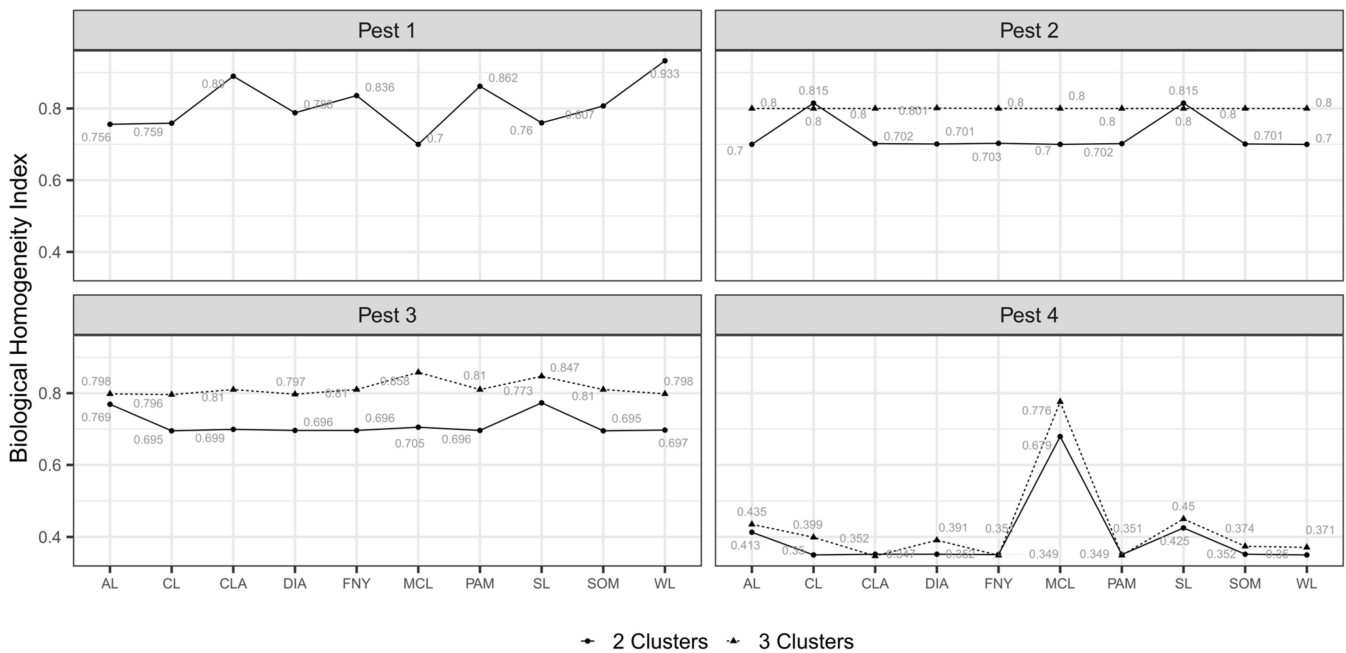

**Figure 8.** Biological homogeneity index (BHI) values calculated for complementary partition essays implemented over absence-only clusters of binary models. This index bases partition selection on the highest observed values.

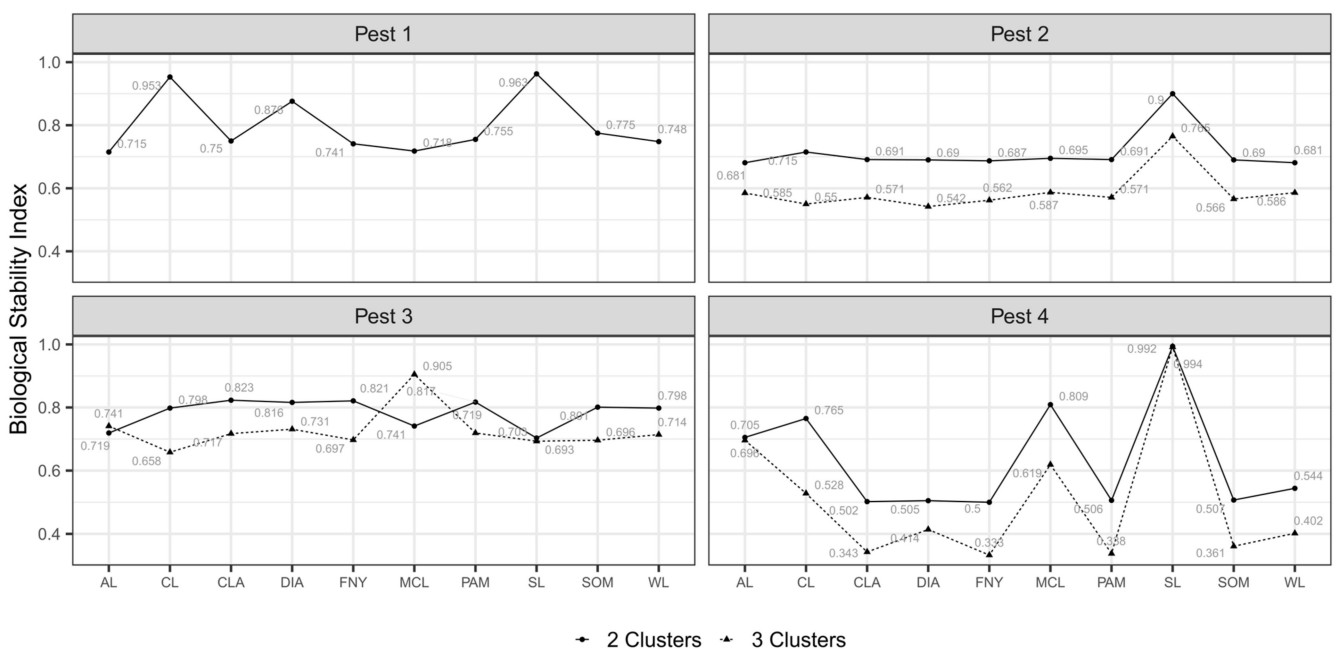

**Figure 9.** Biological stability index (BSI) values calculated for complementary partition essays implemented over absence-only clusters of binary models. This index bases partition selection on the highest observed values.

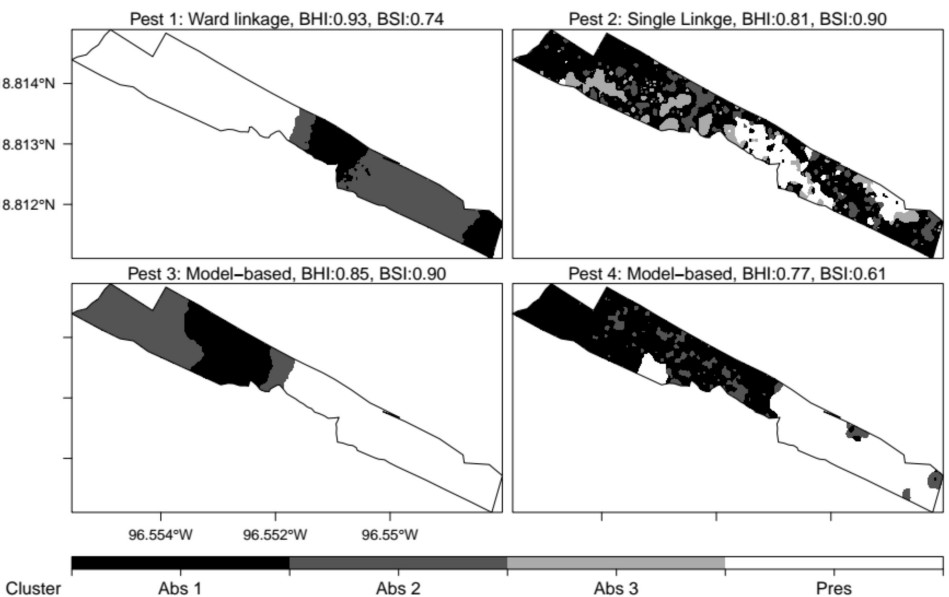

**Figure 10.** Best partition models of absence-only clusters used to isolate nested presences and absences from inadequate parent clusters.

### 3.4. Classified Management Zones

The preliminary field partition model generated for pest 1 showed good visual agreement with its corresponding distribution map (Figure 11, left). Moreover, only one absence zone was recognized (i.e., zone 1, suitability: 0.23), as well as three zones with differentiated levels of pest presence (i.e., zone 2, suitability: 0.41; zone 3, suitability: 0.43; zone 4, suitability: 0.59). This partition model also showed statistically supported environmental agreement with presence and absence zones included in the reclassified distribution model generated for pest 1. In the case of pest 2 (Figure 11, right), visual agreement between its preliminary field partition model and its corresponding reclassified distribution model was also good but partial, since a significant area of the orchard where the target pest was known to be present (roughly one-quarter of the total field) was associated with suitability values below the established threshold for presence-absence discrimination (i.e., 0.25). Three absence zones were identified (i.e., zone 1, suitability: 0.15; zone 2, suitability: 0.15; zone 4, suitability: 0.16) as well as three more zones of differentiated pest presence (i.e., zone 3, suitability: 0.41; zone 5: suitability: 0.46; zone 6, suitability: 0.57). Both sets of zones displayed high environmental agreement with true presence and absence zones included in the reclassified distribution model generated for pest 2.

The preliminary partition model generated for pest 3 (Figure 12, left) showed good visual and environmental agreement with its corresponding reclassified distribution model. Only one pest absence zone was recognized (suitability: 0.25), although a portion of it was incorrectly included in a pest presence management zone (i.e., zone 3). The highest mean suitability was observed in zone 2 (0.71), whereas those of zones 3 and 4 ranged from 0.55 to 0.64. In the case of pest 4 (Figure 12, right), the generated field partition model showed good agreement (both environmental and visual) with its corresponding reclassified pest distribution map; nevertheless, none of the delineated MZ could be classified as absence-only. Instead, four pest presence levels were recognized (zone 1, suitability: 0.32; zone 2, suitability: 0.45; zone 3, suitability: 0.45, zone 4, suitability: 0.67), two of which displayed identical mean suitability values (MZ 2 and 3) but significantly differentiated environmental conditions.

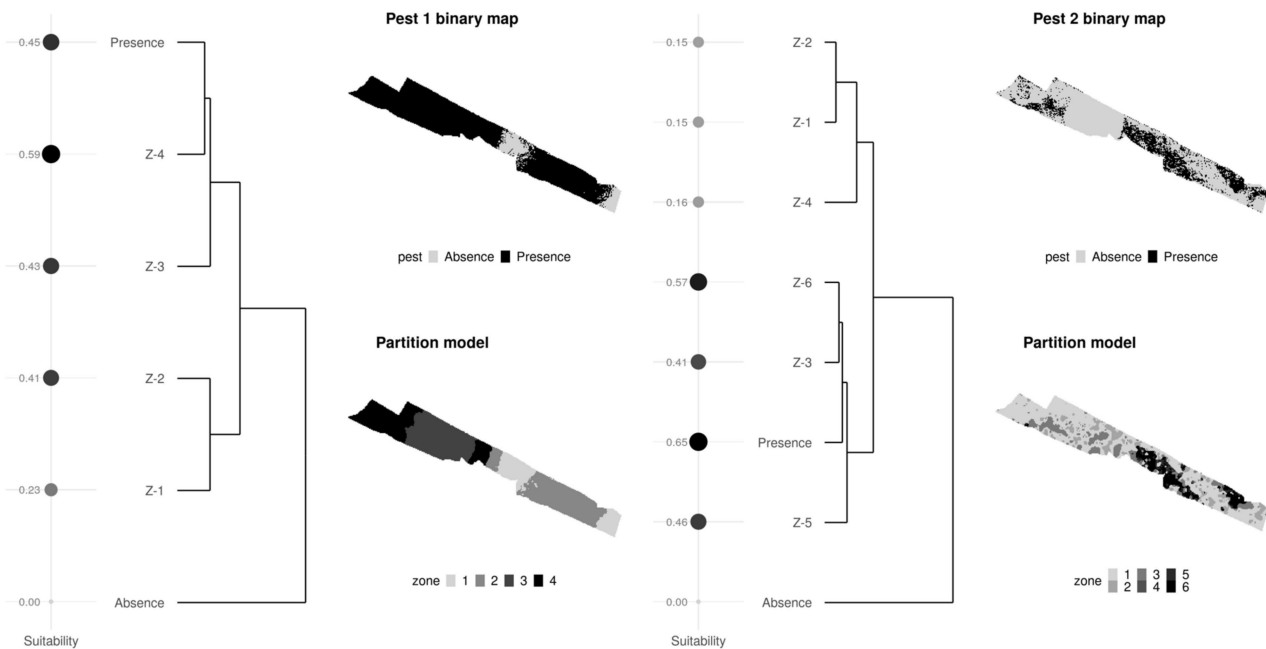

**Figure 11.** Multivariate chart used to classify MZ included in partition models that facilitate within-field management of pests 1 and 2. Environmental dendrogram based on Euclidean distances and average linkage.

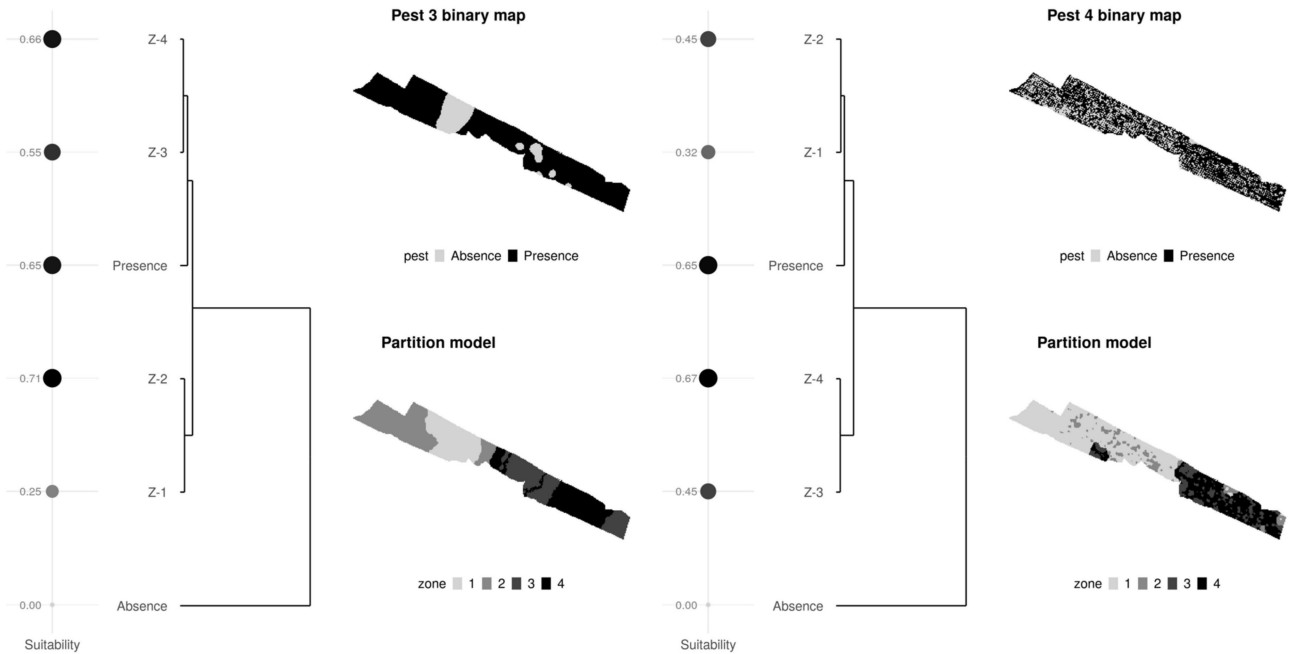

**Figure 12.** Multivariate chart used to classify MZ included in partition models that facilitate within-field management of pests 3 and 4. Environmental dendrogram based on Euclidean distances and average linkage.

Preliminary field partition models generated for pests 5 and 6 displayed good agreement with their corresponding reclassified pest maps. In the case of pest 5 (Figure 13, left), six MZ were delineated within the experimental orchard, with zones 1 through 4 showing mean suitability values below the presence-absence threshold previously established (suitability: 0.0). In an exceptional scenario, MZ that corresponded with known presences of pest 5 (MZ 5 and 6) also showed mean suitability values considerably below

the presence-absence threshold (0.03 and 0.07 respectively), apparently as a result of natural distributional features of pest weeds. The field partition model developed for pest 6 (Figure 13, right) included three MZ, one pest absence zone (zone 1, suitability: 0.08) and two differentiated levels of presence (zone 2, suitability: 0.30; zone 3, suitability: 0.38).

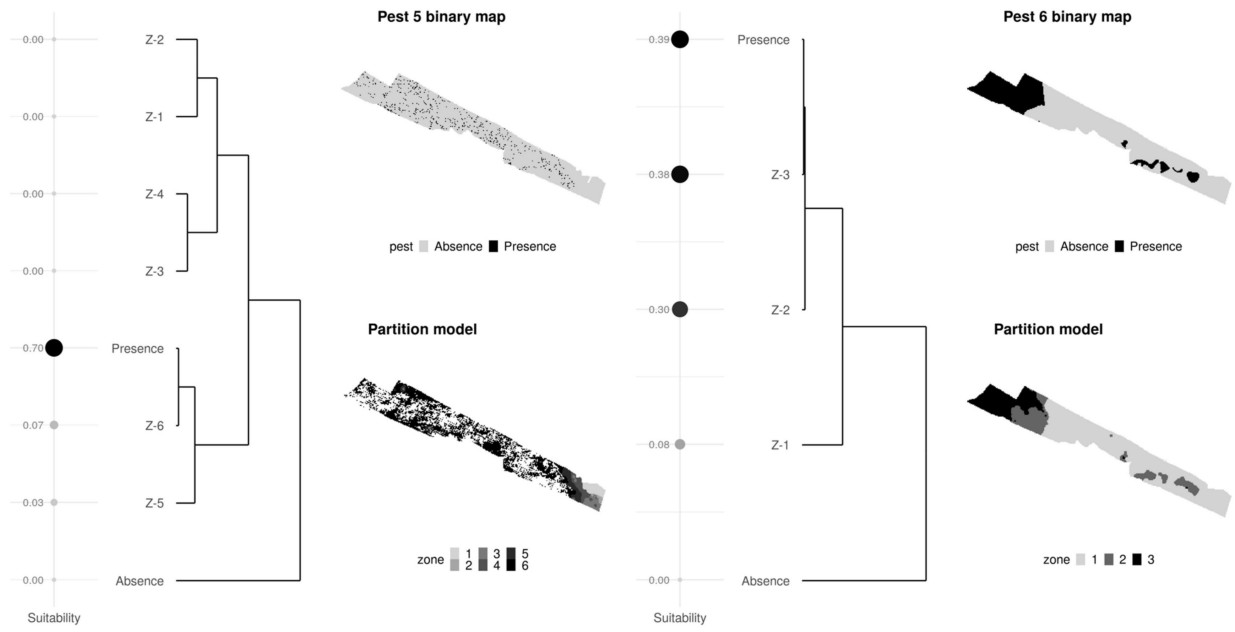

**Figure 13.** Multivariate chart used to classify MZ included in partition models that facilitate within-field management of pests 5 and 6. Environmental dendrogram based on Euclidean distances and average linkage.

Corrected versions of preliminary partition models (i.e., zone number according to mean suitability values, fused redundant zones) generated for all evaluated pests is presented in Figure 14.

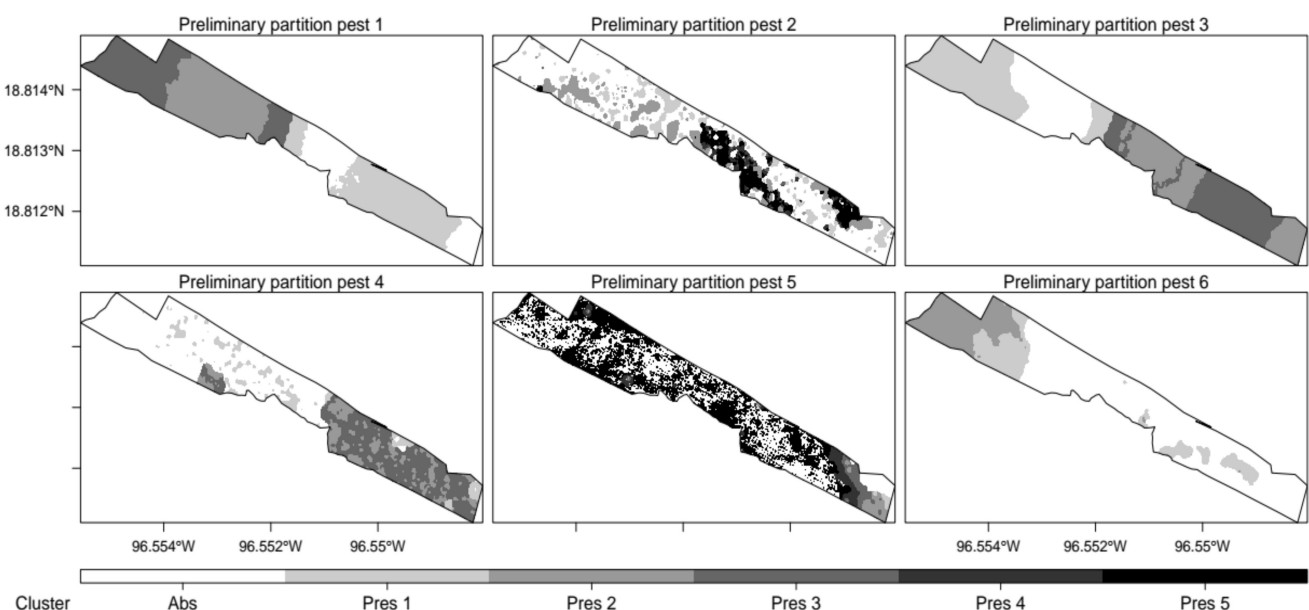

**Figure 14.** Preliminary models of field partition generated to facilitate management of all six virtual pests within the experimental orchard.

### 3.5. Validated Management Zones

For all six virtual pests, networks of environmental relationships (Figure 15) showed varying degrees of similarity between the MZ included in their corresponding field partition models, from no environmental relationships at all ($D = 0$) to completely overlapping environments ($D = 1$). However, significance values associated with such between-zone similarities (p$D$) support only three environmental links that show a maximum $D$ value of 0.02 (thick lines between nodes, pests 2 and 4), which is insignificant in terms of the threshold value used to define strong environmental relationships between MZ ($D \geq 0.1$). Such a condition was interpreted as evidence of environmental independence between the MZ included in preliminary field partition models generated for all six virtual pests. This is important since zones with individual potential to host pest populations are expected to be environmentally independent from the rest [8].

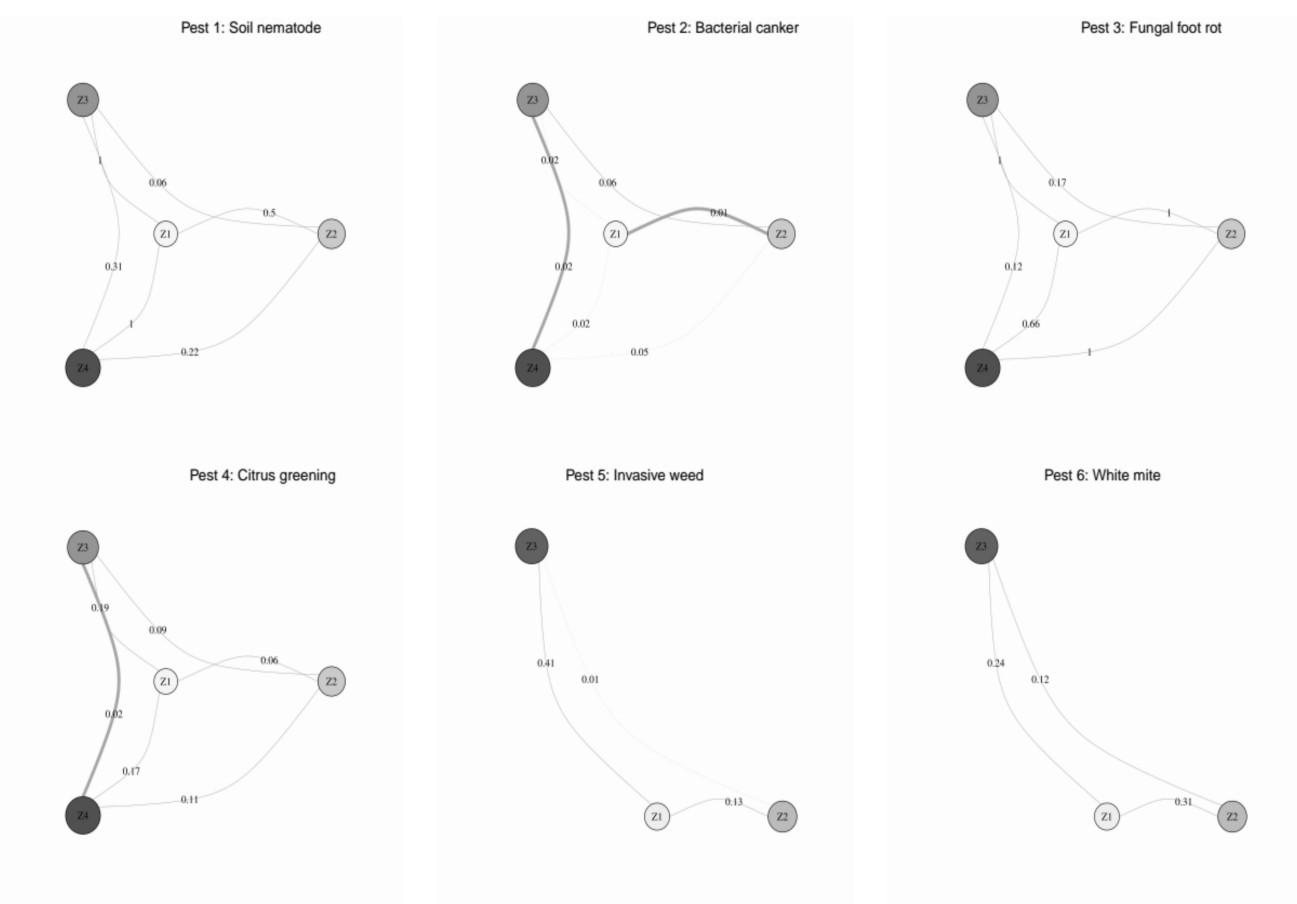

**Figure 15.** Environmental relations between management zones (by pest, 1–6). Zones are represented by nodes of a different color (corresponding to colors used to represent management zones during previous analyses) and size, with bigger nodes representing higher mean suitability values. Environmental distances between MZ (1-$D$) are represented by numbers next to network edges (i.e., links). Link width (i.e., slim, normal, thick) corresponds with the three manifestations of environmental relationships between MZ considered here. Slim edges (barely visible) represent statistically insignificant relationships which showed $D$ values below the established threshold for recognition of strong environmental ties. Normal edges (visible but slim) represent statistically insignificant relationships where $D$ values surpassed the established threshold for recognizing environmental bounds between zones. Thick edges represent statistically significant relationships where $D$ values may or may not have surpassed the threshold established for the recognition of environmental bounds between zones.

Although between-zone overlaps observed in the generated networks did not justify the fusion of MZ for any of the analyzed field partition models, such similarities exist and should be considered when using such models to program/implement pest control practices within the experimental orchard. Final field partition models are presented in Figure 16.

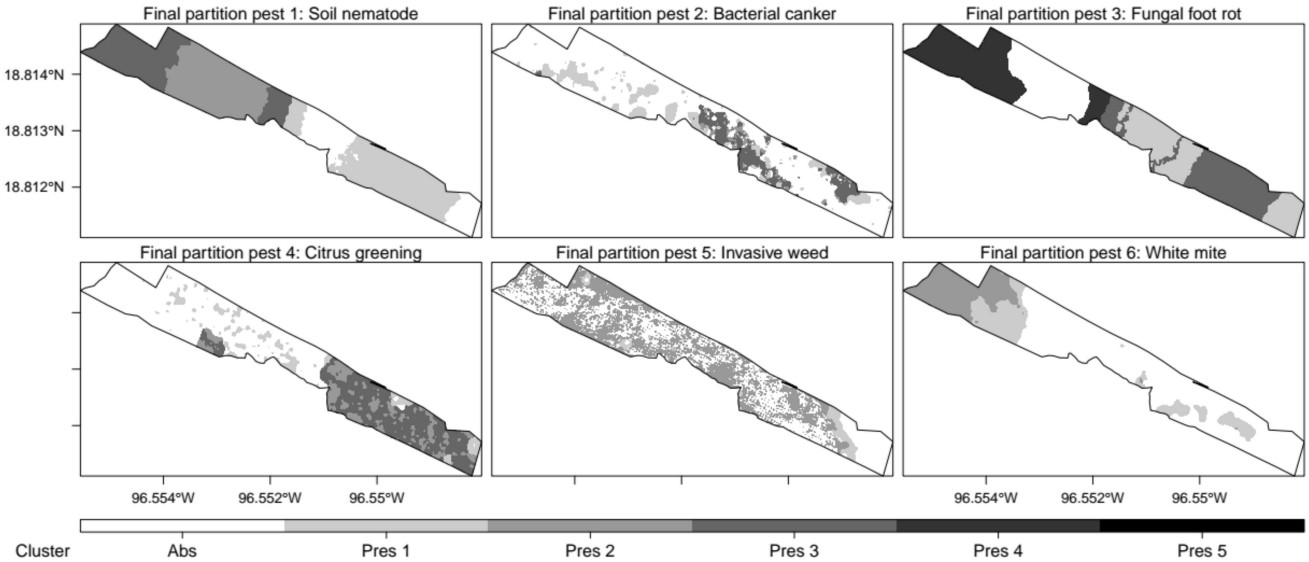

**Figure 16.** Final partition models generated to optimize management of virtual pests within the experimental orchard.

## 4. Discussion

### 4.1. Nested Field Partitioning Essays in eSSPM

Field partitioning strategies currently described in eSSPM are based on one-time implementations of MC algorithms capable of delineating homogeneous and environmentally independent MZ with individual potential to host pest populations [8]. Although this is an efficient approach to partition within-field variability for temporally and spatially stable agricultural phenomena (i.e., yield properties, soil conditions) [61–64], it shows clear limitations when used to partition the spatial variability of agricultural pests which are dynamic in space and time and tend to be present only in specific areas of crop fields (i.e., pest presence zones). Therefore, previous implementations of this method report field partition models with sub-optimal topologies such as presence zones nested within absence clusters, presence zones insensitive to differentiated levels of pest infestation, and more than one pest absence zones [8].

The fact that field partition models generated during the development of this work showed a low propensity to present the inconsistencies mentioned above, was interpreted as evidence that nested partitioning essays are an efficient strategy to minimize the frequency of sub-optimal results during the delineation of MZ with pest control purposes. However, it is necessary to stress the weaknesses that restrain us from claiming universal usefulness for the methods proposed in this paper. In this sense, three types of sub-optimal scenarios were observed: (1) partition models which included nested pest presences in small portions of the crop field (pests 2 and 3), (2) partition models which showed marginal suitability values even in pest presence zones (pest 5), and (3) partition models which showed a pest absence zone considerably larger than expected (pest 4).

For the first scenario, discussions should revolve around the nature of nested field partitioning essays themselves, since even though sub-optimal results in the generated field partition models (i.e., presence zones within absence clusters) were limited to small parts of the crop field, they still exert an influence over the final topologies of such models. This indicates that the two-step process proposed in this paper could be modified to an *n*-step

process which includes as many within-cluster partitioning essays as necessary to isolate nested pest presence zones. In practice, such *n*-step nested partitioning essays should follow the same logic as their two-step predecessors, with biologically meaningful CVI being used to determine optimal MC algorithms to be used and the appropriate number of MZ to be delineated. However, it should be noticed that this improvement of the zoning method originally proposed will imply more processing time although computational needs will remain the same.

For scenarios two and three, discussions should focus on the type of pests intended to be managed. In these cases, sub-optimal zoning results were observed in field partition models generated for citrus greening (pest 4) and invasive weeds (pests 5) specifically. These two pests share NDVI (pest 4) or NDVI by-products (pest 5) as environmental predictors, which preconditions the crop field to show pest presence sites only where some form of vegetation is also present. However, field partitioning essays for these pests included values of all pixels representative of the experimental orchard. This form of implementation seems to have created a special condition where environmental values from places where no vegetation was present combined with values of complementary pest predictors (i.e., soil PH for pest 4, ambient temperature for pest 5) created false zones of marginal pest presence, which acted as confusion factors during the partitioning of the target crop filed. Based on these observations, we recommend that when the pest to be managed strictly needs the presence of vegetation to manifest, predictors to be used during partitioning essays should include a preprocessing that excludes all environmental values occurring in pixels where such a precondition is not fulfilled.

### 4.2. Performance of MC Algorithms within the Context of eSSPM

Although most PA zoning essays reported by literature are based on implementations of particular clustering approaches (e.g., fuzzy c-means, FANNY, McQuitty) [10,11], results presented here show that more than one MC algorithms tend to be necessary to delineate MZ with pest control purposes. This is because the spatial nature of data sets to be clustered during field partitioning essays could (and many times do) vary considerably depending on the pest to be managed and on the portion of the crop field being partitioned. Thus, clustering methods of proven efficiency to develop binary field partitions do not necessarily correspond to those that offer the best performance during complementary ones (whether presence-only or absence-only), even when the same pest is being considered. Similarly, any set of MC algorithms used to partition a crop field assuming a pest "A" will rarely perform accurately during partitioning essays developed for a pest "B."

Despite this result, a general pattern was observed where partitioning essays intended to distinguish between pest presence and pest absence sites (i.e., binary, absence-only) were better resolved by hierarchical MC algorithms (i.e., pests 1, 3, 4, 5, and 6), whereas more sophisticated approaches (i.e., SOM, DIA, MCL) were necessary to find differentiated levels of pest presence within general presence clusters (i.e., pests 1, 4, 5 and 6). This trend is consistent with published research that highlights the efficiency of hierarchical MC algorithms (i.e., SL, CL, WL) to partition sets of well-separated binary data (whether biologically meaningful or not) [10,65,66], as well as that of partitioning and model-based algorithms (i.e., MCL, DIA, SOM) to perform this same task with data sets that include observations more closely positioned in statistical space [50,67,68]. It must be noted, however, that plenty of other research works successfully explore the implementation of hierarchical clustering methods to partition spatially close data sets, as well as partitioning and model-based approaches to partition well separated binary data sets. Taking these observations into account, we recommend developing eSSPM field partitioning essays (i.e., binary, complementary) based on a combination of MC algorithms that best suit the particularities of the specific pests and fields to be managed.

*4.3. Validation of Field Partition Models Using Biologically Meaningful CVI*

A high proportion of field partitioning methods currently described in PA determine optimal numbers of management zones by means of internal and stability CVI [11,37,61,62,64,69], which are specialized algorithms designed to validate unsupervised clustering essays [70]. Internal validation uses intrinsic information in the data to assess the quality of clustering (e.g., compactness, contentedness, fuzziness performance, partition entropy, fuzziness performance), whereas stability measures evaluate the consistency of a clustering topology by comparing results from different iterations where each column (i.e., observation) is removed one at a time during the clustering process (e.g., average proportion of non-overlap, average distance, average distance between means) [50]. It is necessary to consider, however, that these examples do not deal with the delineation of management zones with pest management purposes. Instead, they focus on the management of resources such as fertilizers and water which are inputs needed when soil conditions are not favorable for plant fertility.

The task of enclosing pest populations within environmentally homogeneous zones poses different technical complexities than enclosing qualitative classes of the crop itself. For instance, zoning methods used in fertilization and irrigation PA are based on environmental features that are more stable in time (e.g., soil properties, terrain form) and agricultural phenomena that are driven almost exclusively by such factors (e.g., soil fertility, water deficit). On the other hand, in the case of eSSPM it is necessary to consider that species are living entities constantly evolving to occupy as many environments as possible within the boundaries of their physiological limitations (i.e., adaptation [71]). This means that they will rarely restrict their natural expansion to the limits of a single "climatic component" (i.e., set of environmental conditions) as demonstrated by ecological and agronomic literature [72–75]. On the contrary, species tend to occupy different climatic components simultaneously, depending on factors such as the time of the year and the immediate needs of their populations [76].

Since agroecosystems follow the same physical rules as natural ecosystems (at least in nature), the existence of differentiated microenvironmental conditions can be assumed for most agricultural fields [77,78]. We can also assume that within-field pest distribution will usually converge with more than one microenvironmental component, and that such components are not the only factors governing how pests distribute within the crop field but complementary influences that closely interact with other pest drivers such as the available resources (e.g., food abundance, mating sites) [79]. This is the main reason why we find internal and stability CVI lacking as validators of field partitioning essays in eSSPM, because of their natural tendency to favor partition models that maximize internal coherence of microclimatic components (i.e., cluster) that, although relevant, are not the only drivers of pest within-field dynamics.

BHI compensates for the influence of unknown pest drivers by seeking maximum congruence between microenvironmental components that are somewhat homogeneous in nature (i.e., clusters) and the known spatial distribution of target pests. This way it is possible to evaluate field partitioning essays in function of clusters' capability to enclose individuals of the same class (e.g., levels of pest infestation) rather than their internal structure (which excludes the influence of other factors but microenvironmental). Despite these encouraging conclusions, there are limitations in the use of BHI that need to be mentioned. For instance, when presence-absence thresholds were set too low or too high during binary field partitioning essays, BHI was unable to identify best performing partition models. Under such circumstances, BHI gave higher scores to partition models that were composed of one big cluster containing most observations and a second much smaller cluster that included few isolated values. All these models showed BHI values near to 1.

The reason for the observed phenomenon is that, at least in great measure, original implementations of BHI (and BSI) were designed to be used with genes instead of suitability classes [56]. Gene classes make sense regardless of them forming part of big or small clusters, since they all have a specific function. In our case, however, functional classes

were represented by groups of places with similar potential to host pest populations (i.e., MZ). Since this potential is not an intrinsic attribute of the pest itself but of the geographic space occupied by it (i.e., distribution area), its gradation into functional intervals (i.e., pest levels) represents a rather subjective task with more than one equally plausible outcomes. This created scenarios where, even when the same MC algorithm was implemented with the same data sets, using different threshold values to define pest levels resulted in partition models with varying degrees of dissimilarity with ground truth samples (i.e., virtual data).

Although BSI was useful as a secondary validator when different models showed the highest BHI, it did not offer any means to facilitate the validation process in cases where bad model partitions showed high BHI values. To facilitate the recognition of these suboptimal partition models, we recommend the exploration of $S^2_T$ or "total within-zone pest suitability variance". $S^2_T$ was designed to validate field partitioning essays with crop management purposes [36] and recently adapted to the needs of site-specific pest management [8]. Since higher $S^2_T$ values indicate more heterogeneous classes (i.e., MZ), useful field partition models should show low scores for this index. This way, when a model shows extremely high BHI and an extremely low $S^2_T$ values, it should be regarded with caution since suitability thresholds used to define pest levels might still need proper tuning.

### 4.4. SDM-Based Validation of Management Zones

As mentioned before, once a crop field has been subdivided into individual MZ, it is necessary to corroborate the existence of differences between them in terms of the agricultural phenomenon to be managed. This process, known as validation of management zones, can be developed through different statistical approaches such as clustering validation indexes. CVI represent the most cited approach to the validation of MZ in PA; nevertheless, there are numerous indexes available for such purposes today and no clear agreement in terms of which one offers the best results [10]. Moreover, they are incapable of assessing agronomically meaningful differences between MZ. ANOVA tests, on the other hand, are a straightforward means to corroborate the existence of statistically significant differences between MZ. However, they are limited to the evaluation of individual variables that could or could not reflect the multidimensionality of complex environments. Additionally, they assume independence in the input dataset, a condition that is not met when environmental values are spatially referenced [11]. Finally, although MLM do account for spatial correlation in the data and consider the conjunct effect of different variables over the modeled phenomenon, they demand meticulous parameterization and their implementation tends to be more computationally intensive than other methods [11].

The validation of MZ based on SDM background tests offers different advantages. Since SDM tools (e.g., background tests) were developed to facilitate the study of species geographic distributions, they are spatially explicit in design. This is relevant because they offer *ad hoc* methodologies to minimize the effect of spatial biases usually present in SDM-related processes such as modeling species distributions and comparing environmental preferences between species/populations (e.g., differences in sampling effort between populations, spatially uneven samplings, differences in the habitat available to populations in geographic regions where they do not overlap) [58]. Moreover, SDM tools perform between-zone comparisons in terms of complex multivariate environments and determine environmental similarity based on relative rather than absolute measures (i.e., Schoener's *D*) [59]. These features make SDM background tests a more realistic and flexible approach to validating MZ than the rest of methodologies discussed above (at least in SSPM). Finally, the use of ecological networks to explain inter-zonal environmental differences is not only easy to set up (only two parameters are needed; i.e, a *D* threshold and a significance level for such a threshold) but also improves considerably the interpretability of SDM background tests when used to validate site-specific management zones with pest control purposes.

*4.5. New Workflow for MZ Delineation in eSSPM*

Based on the results and discussions presented during the development of this work, an improved version of current eSSPM zoning methods was described. This new workflow consists of five straightforward steps: (1) description of cause–effect relationships between georeferenced presence–absence data and mapped environmental predictors (not discussed in this paper), (2) single binary partition of a crop field validated through biologically meaningful CVI, (3) *n* complementary partitions of presence-only and absence-only clusters validated through biologically meaningful CVI, (4) suitability-based classification of MZ, and (5) validation of MZ based on SDM background tests.

## 5. Conclusions

Results generated in this work support the fact that delineating pest management zones (MZ) based on nested field partitions of environmental features represents an effective means to overcome many of the limitations associated with zoning methods previously described in eSSPM, such as presence zones insensitive to differentiated levels of infestation and nested presences/absence. In general, more than one MC algorithm is necessary to delineate accurate MZ in eSSPM. Hierarchical MC algorithms tended to generate better outputs during binary and absence-only partitioning essays, whereas more sophisticated approaches (i.e., model-based, machine learning) tended to outperform the rest during presence-only complementary partitions.

The validation of partitioning essays based on biologically meaningful CVI (i.e., BHI, BSI) are of great utility during the selection of best-performing algorithms and optimum numbers of MZ to be delineated. Still, due to inaccuracies observed in specific circumstances, we recommend complementing the evaluation of field partitioning essays with $S^2_T$. We also conclude that the visual aids used during the classification of MZ (e.g., dendrograms, bubble charts, maps) are important resources that facilitate relevant agronomic decisions such as what zones should be considered individual subfield units and what zones should be fused together. Similarly, SDM background tests displayed as ecological networks represent an efficient and flexible way to corroborate the environmental uniqueness of MZ and corroborate cases of fusion/separation of zones.

A novel workflow to the delineation of MZ within the context of eSSPM was described. It is based on the following sequential steps: (a) selection and mapping of zoning factors, (b) binary partition of the managed crop field, (c) complementary partitions (i.e., presence-only, absence-only) of the managed crop field, (d) classification of zones included in preliminary partition models, and (e) ecological validation of corrected (i.e., final) zones.

**Author Contributions:** Conceptualization, L.J.M.-V.; writing—original draft preparation, L.J.M.-V. and A.L.-N.; writing—review and editing, L.J.M.-V., A.L.-N., R.L.-C. and S.C.-E.; visualization, L.J.M.-V., A.L.-N., R.L.-C., S.C.-E.; supervision, A.L.-N.; funding acquisition, A.L.-N. and L.J.M.-V. All authors have read and agreed to the published version of the manuscript.

**Funding:** This research received funding from Fideicomiso o Fondo institucional de Fomento Regional para el Desarrollo Científico, Tecnológico y de Innovación (FORDECyT)-Consejo Nacional de Ciencia y Tecnología (CONACyT) project "Generación de estrategias científico-tecnológicas con un enfoque multidisciplinario e interinstitucional para afrontar la amenaza que representan los complejos ambrosiales en los sectores agrícola y forestal de México" [292399, 2018].

**Institutional Review Board Statement:** Not applicable.

**Informed Consent Statement:** Not applicable.

**Acknowledgments:** We thank Simón del Valle for facilitating unlimited access to his commercial orchard during the development of this research, as well as to the Méndez-Vázquez family for hosting our work team indefinite time during the development of this investigation. We also thank Luis Alberto Sánchez-Tolentino (R.I.P.) and Guadalupe Chávez-Hidalgo for their unvaluable help during fieldwork and laboratory analyses. L.J.M.-V. received scholarship support from the Mexican Consejo Nacional de Ciencia y Tecnología (CONACyT) to develop this project as part of his graduate studies.

**Conflicts of Interest:** The authors declare no conflict of interest.

**Appendix A**

A list with the abbreviations and acronyms referred to in this work are presented in Table A1 to facilitate further review.

**Table A1.** Meanings of abbreviations and acronyms referred to in this work.

| Abbreviation | Meaning | Class |
|---|---|---|
| AL | average linkage | clustering algorithm |
| CL | complete linkage | clustering algorithm |
| CLA | clustering large applications | clustering algorithm |
| DIA | divisive analysis | clustering algorithm |
| FNY | fuzzy analysis | clustering algorithm |
| MCL | model-based clustering | clustering algorithm |
| PAM | partitioning around medioids | clustering algorithm |
| SL | single linkage | clustering algorithm |
| SOM | self-organizing maps | clustering algorithm |
| WL | Ward's linkage | clustering algorithm |
| SSPM | site-specific pest management | discipline |
| eSSPM | ecological site-specific pest management | discipline |
| IPM | integrated pest management | discipline |
| PA | precision agriculture | discipline |
| SDM | species distribution modeling | discipline |
| SSIPM | site-specific insect pest management | discipline |
| aFRot | active foot rot | pest driver |
| cropFVC | fractional vegetation cover of the research orchard | pest driver |
| cropHeight | height of trees included in the research orchard | pest driver |
| cropNDVI | normalized differences vegetation index of the research orchard | pest driver |
| DSM | digital surface model | pest driver |
| DTM | digital terrain model | pest driver |
| flowAccum | flow accumulation | pest driver |
| flowDir | flow direction | pest driver |
| FVC | fractional vegetation cover | pest driver |
| iFRot | inactive foot rot | pest driver |
| maxTemp | maximum ambient temperature | pest driver |
| minTemp | minimum ambient temperature | pest driver |
| NDVI | normalized differences vegetation index | pest driver |
| relHum | relative humidity | pest driver |
| SI-NDVI | single-image normalized differences vegetation index | pest driver |
| soilEC | soil electrical conductivity | pest driver |
| soilPH | soil potential of hydrogen | pest driver |
| sunRad | sun radiation | pest driver |
| TDS | total dissolved solids | pest driver |
| TRI | topographic roughness index | pest driver |
| VPD | vapor pressure deficit | pest driver |
| FPS | frames per second | precision agriculture tool |
| GIS | geographic information system | precision agriculture tool |
| GPS | global positioning system | precision agriculture tool |
| MZ | management zones | precision agriculture tool |
| UAS | unmanned aerial system | precision agriculture tool |
| ANOVA | analysis of variance | statistical method |
| BHI | biological homogeneity index | statistical method |
| BSI | biological stability index | statistical method |
| CVI | classification validation index | statistical method |
| $D$ | Schoener's $D$ | statistical method |

**Table A1.** *Cont.*

| Abbreviation | Meaning | Class |
|---|---|---|
| IDW | inverse distance weights | statistical method |
| MC | multivariate clustering (algorithm) | statistical method |
| MLM | mixed linear models | statistical method |
| PAST | presence–absence suitability threshold | statistical method |
| $p$D | probability of $D$ | statistical method |
| $S^2_T$ | total within-field suitability variance | statistical method |

Four types of MC algorithms were used during the development of this work: hierarchical, partitioning, machine learning (based), and model based. Hierarchical clustering algorithms are based on agglomerative methods that yield a dendrogram which can be cut at a chosen height to produce the desired number of clusters [50]. Each observation is initially placed in its own cluster and the clusters are successively joined together in order of their "closeness". The closeness of any two clusters is determined by a dissimilarity matrix and can be based on a variety of agglomeration methods, which in the case on this work were:

1. Average linkage [51], mean distance between observations.
2. Complete linkage [51], maximum distance between observations.
3. Single linkage [51], minimum distance between observations.
4. Ward linkage [55], error sum of squares.

For all cases, the manipulation of the distance function exerts an influence on the combination of any two groups to form a new one [10]. Manipulation of such a distance function can be achieved through the equation:

$$D\big(G_x, \big(G_i, G_j\big)\big) = \alpha_i D(G_x, G_i) + \alpha_j D\big(G_x, G_j\big) + \beta D\big(G_i, G_j\big) + \mu \big|D(G_x, G_i) - D\big(G_x, G_j\big)\big|, \tag{A1}$$

where $D$ is a distance function, $\alpha_i$, $\alpha_j$, $\beta$ and $\mu$ are coefficients that have their values determined according to the applied algorithm.

A fifth hierarchical clustering approach was tested in this work:

1. DIANA (Divisive analysis [52]) is an algorithm that initially starts with all observations in a single cluster, and successively divides the clusters until each one contains a single observation; thus, hierarchies are built in $n - 1$ steps. During each step, the cluster $C$ with the largest diameter is selected based on the following equation:

$$diam(C) := max_{i,j \in C} d(i, j) \tag{A2}$$

Assuming $diam(C) > 0$, we then split up $C$ into two clusters $A$ and $B$, according to a variant of the method of Macnaughton-Smith et al. [80]. At first $A := C$ and $B := \theta$, later one object is moved from $A$ to $B$ and then other objects are moved from $A$ to $B$.

Partitioning algorithms divide a set of elements into $k$ groups without constructing a hierarchical structure, following the principle that elements in a same group should be more similar than elements belonging to different groups [50]. These algorithms perform a division of the data to identify $n$ natural groups into a certain number of disjoint groups, with a centroid for each group as a reference and employing a distance function. They can perform clustering automatically and seek to achieve the maximum similarity between the elements of the same group and the minimum similarity between different groups [10]. In our case, the following algorithms were used to implement partitioning clustering during the development of this work:

2. PAM (Partitioning around medioids [52]), similar to "k-means", the number of clusters (i.e., $k$) is fixed in advance and an initial set of cluster centers (i.e., "medioids", in contrast to "means" used in k-means) is required to start the algorithm. PAM is considered more robust than k-means because it admits the use of other dissimilarities

besides Euclidean distance. The implementation of PAM clustering was based on the equation:

$$TD := \sum_{i=1}^{k} \sum_{x_j \in C_i} d(x_j, m_i), \qquad (A3)$$

where $TD$ is the total deviation, defined as the sum of dissimilarities of each point $X_j \in C_1$ to medioid $m_i$ of its cluster.

3. CLARA (Clustering large applications [52]), a sampling-based algorithm that implements PAM on a number of sub-datasets, which allows for faster running times when a number of observations is relatively large. CLARA complies with the following algorithm:

    a. Create randomly, from the original dataset, multiple subsets with fixed size (sampsize).

    b. Compute PAM algorithm on each subset and choose the corresponding k representative objects (medioids). Assign each observation of the entire data set to the closest medioid.

    c. Calculate the mean (or the sum) of the dissimilarities of the observations to their closest medioid. This is used as a measure of the goodness of the clustering.

    d. Retain the sub-dataset for which the mean (or sum) is minimal. A further analysis is carried out on the final partition.

4. FANNY (Fuzzy analysis [52]), this algorithm performs fuzzy clustering, where each observation can have partial membership in each cluster. Thus, each observation has a vector that gives the partial membership to each of the clusters. A hard cluster can be produced by assigning each observation to the cluster where it has the highest membership. FANNY clustering is based on the equation:

$$C = \sum_{v=1}^{k} \frac{\sum \sum u_{iv}^r u_{jv}^r d(i,j)}{2 \sum u_{jv}^r}, \qquad (A4)$$

where $u_{iv}$ is the membership of element $i$ in relation to group $v$, $n$ is the number of elements that form the data set, $k$ is the number of groups to be formed, $r$ corresponds to a pertinent exponent, and $d(i,j)$ is the distance between elements $i$ and $j$.

Machine learning algorithms possess the capability of improving their performance automatically through experience. To do so, they build a mathematical model based on sample data, known as "training data", in order to make predictions or decisions without being explicitly programmed to do so. Although machine learning approaches are commonly associated to modeling and prediction tasks, these tools can also be used to develop clustering essays [81]. In this work, only one machine learning clustering algorithm was used:

5. SOM (Self-organizing maps [54]), an unsupervised learning technique based on neural networks that is popular among computational biologists and machine learning researchers. SOM is a concept of competition network that tries to find the most similar distance between the input vector and neuron with weight vector $w_i$. SOM always consist of both input vector $x$ and output vector $y$. At the start of the learning, all the weights ($w_i$) are initialized to small random numbers. The set of weights forms a vector $w_i = w_{ij}$, $i = 1, 2, \ldots, k_x$, $j = 1, 2 \ldots, k_y$ where $k_x$ is the row number and $k_y$ is the column number. Euclidian distance $d$ between the input vector $x$ and the neuron with weight vector of the given neuron $w_c$ is computed by:

$$d(x, w) = |x(t) - w_c(t)|, \qquad (A5)$$

where $t$ is an integer. Next, SOM will search for the winner neuron using the minimum distance (best matching unit, BMU). BMU is calculated as follows:

$$BMU = \mathrm{argmin}\, |x(t) - w_c(t)| \qquad (A6)$$

To increase the similarity with the input vector, weights are adjusted after obtaining the winning neuron. The rule for updating the weight vector is given by:

$$w_i(t+1) = \frac{\sum h_{c(i)}(t) \times x_j}{\sum h_{c(i)}(t)}, \tag{A7}$$

where $w_i(t+1)$ is the updated weight vector, $x_j$ is the input record, $h_c(i)(t)$ is the neighborhood function related to the winning unit $c_i$ at step $t$, and $S$ is the number of input samples. The neighborhood function (usually assumed as Gaussian) determines the rate of change of the neighborhood around the winner neuron as in equation:

$$h_{c(i)}(t) = e^{\frac{-\left|r_{c(i)} - r_i\right|^2}{2\sigma(t)^2}}, \tag{A8}$$

where $r_{c(i)}$ and $r_i$ are, respectively, the positions on the map of the winning neuron and of the generic unit $i$; $\sigma(t)$ is the neighborhood radius at the iteration $t$ of the training process and corresponds to the width of the neighborhood function at step $t$. Initially, $\sigma(t)$ can be as large as the size of the map and then, to guarantee convergence and stability, it decreases linearly with time till one during the process.

Finally, model-based clustering algorithms are those that postulate a generative statistical model for the data and then use a likelihood (or posterior probability) derived from this model as the criterion to be optimized. Model-based clustering has recently gained widespread use both for continuous and discrete domains mainly because it allows one to identify clusters based on their shape and structure rather than on proximity between data points [82]. One model-based clustering algorithm was considered in this work:

6.  MCL (Model-based clustering [53]) operates on the assumption that the analyzed data originate from a finite mixture of underlying probability distributions [83]. Each mixture component represents a cluster, and the mixture components and group memberships are estimated using maximum likelihood (EM algorithm). MCL usually assumes a normal or Gaussian mixture model as in the following equation:

$$\prod_{i=1}^{n} \sum_{k=1}^{G} \tau_k \varnothing_k(x_i | \mu_k, \Sigma_k), \tag{A9}$$

where $G$ is the number of components, $x$ represents the data, $\varnothing_k$ are the density and parameters of the $k^{\text{th}}$ component in the mixture, $\mu_k$ (mean vector) and $\Sigma_k$ (covariance matrix) are parameters to model each component $k$ by the multivariate distribution, $\tau_k$ is the probability that an observation belongs to the $k^{\text{th}}$ component, and:

$$\varnothing_k(x_i | \mu_k, \Sigma_k) = (2\pi)^{-p/2} |\Sigma_k|^{-1/2} exp\left\{ -\frac{1}{2}(x_i - \mu_k)^T \Sigma_k^1 (x_i - \mu_k) \right\}. \tag{A10}$$

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
