# Peer review of "Using Simulated Pest Models and Biological Clustering Validation to Improve Zoning Methods in Site-Specific Pest Management"

_applsci, doi:10.3390/app12041900_

Round 1

Reviewer 1 Report

The manuscript presents simulated pest models and biological clustering validation to improve zoning methods in site-specific pest management. The subject is interesting and the contents are easily comprehensible, very well written and organized. I unreservedly recommend acceptance of this manuscript.

Author Response

We appreciate the positive comments on our work from this reviewer, as well as the valuable suggestions from reviewers 2 and 3 that helped improve the original version of our manuscript. You should be able to check the responses to the suggestions from reviewers 2 and 3 in their own section.

Reviewer 2 Report

The paper entitled “Using simulated pest models and biological clustering validation to improve zoning methods in site-specific pest management” proposed a framework for improving zoning methods in site-specific pest management (SSPM). The framework aims to evaluate the utility of partitioning essays to improve the accuracy of current SSPM methods. It consists of a two-steps process where a rough distinction between pest presence and pest absence zones precedes the subdivision of resulting sub-field units. The framework uses a biological clustering validation metric to evaluate the performance of 10 clustering algorithms and choose appropriate numbers of management zones during field partitioning essays. The experiment was conducted on a field-level environmental dataset from a Tahiti lime orchard and realistic simulations of six citrus pests. The results have shown that nested partitioning essays outperform previous methods.

In general, the topic invested in this paper is interesting and suitable for the scope of the AS journal. The paper is well written, and the results seem to be reasonable. The Introduction is thorough and covers the latest works in this field. More importantly, the proposed framework was evaluated on a real-life dataset which increases the contributions of the paper to both literature and industry, thus it may gain high interest from readers.

In support of this paper, the authors should revise the paper to further improve its quality before I vote for an acceptance. My comments are as follows

  • In the Introduction, draw a figure of the workflow to illustrate the main idea invented in this paper. I suggest a thorough workflow from the input to the output of the model.
  • Even though I try to read the whole paper but sometimes I need to track back the text to check for the meaning of several abbreviations. Thus, I suggest authors insert a table of notations and abbreviations to enhance the readability of the paper.
  • Authors need to revise all figures from Figs 1 to 9 and in the Appendix since the current forms are not in a good shape. All figures should be clearer, the text is copiable, and not be broken when zooming out.
  • In section 2, revise the equation (1), it should be $ \sum_{k=1}^{K}$.
  • In section 2, I suggest the authors use HDBSCAN on the experimental dataset. HDBSCAN is a hierarchical density-based clustering with the ability to remove noises from the data. In many cases, HDBSCAN outperforms partitioning and model-based clustering.
  • In section 3, I suggest authors use figures or tables to make the comparison between clustering algorithms instead of using plain text.
  • In section 4, although the finding in this paper is supported by the results, the authors should discuss possible methods for this dataset. I still think there are several ways to test the performance of clustering algorithms such as using internal validation metrics. Thus, authors should discuss several options when using clustering algorithms for nested field partitioning essays.  Here are several good examples using HAC authors should refer into the discussion from Applied Sciences [https://doi.org/10.3390/app112311122], [https://doi.org/10.1007/978-981-15-1209-4_1].
  • There are several typos and grammar mistakes in the text, authors should proofread the paper to fix them.

Reviewer 3 Report

This paper used simulated pest models and biological clustering validation to improve zoning methods in SSPM, and it provides a new method for pest control research. Of course, further analysis of the research results is needed, and the conclusions need to be reorganized.

Round 2

Reviewer 2 Report

I have checked this revision. The authors have improved the quality of the paper. Although they declined my several suggestions, they still gave reasonable explanations for those. Therefore, I vote for an acceptance.